# Improving the sensitivity of in vivo CRISPR off-target detection with DISCOVER-Seq+

Roger S. Zou [1,2], Yang Liu[2,3,9], Oscar E. Reyes Gaido [4,9], Maximilian F. Konig[5,9], Brian J. Mog [1,6,7], Leo L. Shen [2], Franklin Aviles-Vazquez[2], Alberto Marin-Gonzalez[2,3] & Taekjip Ha [1,2,3,8] ✉

Discovery of off-target CRISPR–Cas activity in patient-derived cells and animal models is crucial for genome editing applications, but currently exhibits low sensitivity. We demonstrate that inhibition of DNA-dependent protein kinase catalytic subunit accumulates the repair protein MRE11 at CRISPR–Cas-targeted sites, enabling high-sensitivity mapping of off-target sites to positions of MRE11 binding using chromatin immunoprecipitation followed by sequencing. This technique, termed DISCOVER-Seq+, discovered up to fivefold more CRISPR off-target sites in immortalized cell lines, primary human cells and mice compared with previous methods. We demonstrate applicability to ex vivo knock-in of a cancer-directed transgenic T cell receptor in primary human T cells and in vivo adenovirus knock-out of cardiovascular risk gene *PCSK9* in mice. Thus, DISCOVER-Seq+ is, to our knowledge, the most sensitive method to-date for discovering off-target genome editing in vivo.

CRISPR–Cas genome editing is a transformative technology with wide-ranging applications, from interrogating basic biological systems to curing genetic diseases in humans[1]. Genome editing by a CRISPR-associated endonuclease such as *Streptococcus pyogenes* Cas9 relies on the targeted induction of DNA double strand breaks (DSBs), leading to the recruitment of DNA repair factors that repair and potentially modify the genome[2]. However, unintended off-target DNA damage and mutagenesis remain leading concerns for safety and applicability. Therefore, accurate and sensitive methods for discovery of CRISPR–Cas off-target activity are essential[3].

There are numerous methods for detecting off-target CRISPR–Cas activity, but the majority are limited to purified DNA[4–7] or restricted cellular systems such as immortalized cell lines or reporter cells[8–10]. Measurements in these systems may not translate to in vivo applications. For example, Cas9 behavior such as binding kinetics is very different

in vitro[11], and the epigenome, which is highly divergent between different cell types[12], strongly influences CRISPR genome editing activity[13]. Therefore, off-target discovery directly in ex vivo and in vivo model systems is highly desired. However, the few methods compatible with these systems may be constrained by limited sensitivity, due to requiring detection of either transient DNA repair protein binding[14] or of mutations that occur at very low frequencies at some off-target sites[15,16].

In this study, we combined detection of a highly specific DNA repair factor, MRE11 (refs. 14,17,18), with an inhibitor of DNA repair[19] that retains MRE11 residence on genomic DNA, to detect genome-wide CRISPR off-target activity with high sensitivity. Termed DISCOVER-Seq+, this technique enhanced the discovery of CRISPR–Cas-targeted sites in numerous contexts, including in immortalized cell lines, primary human cells and mice at clinically relevant targets. Together, DISCOVER-seq+ represents, to our knowledge, the most

[1]Department of Biomedical Engineering, Johns Hopkins University School of Medicine, Baltimore, MD, USA. [2]Department of Biophysics and Biophysical Chemistry, Johns Hopkins University School of Medicine, Baltimore, MD, USA. [3]Department of Biophysics, Johns Hopkins University, Baltimore, MD, USA. [4]Department of Medicine, Johns Hopkins University School of Medicine, Baltimore, MD, USA. [5]Division of Rheumatology, Department of Medicine, Johns Hopkins University School of Medicine, Baltimore, MD, USA. [6]Ludwig Center, Sidney Kimmel Comprehensive Cancer Center, Johns Hopkins University School of Medicine, Baltimore, MD, USA. [7]Lustgarten Pancreatic Cancer Research Laboratory, Sidney Kimmel Comprehensive Cancer Center, Johns Hopkins University School of Medicine, Baltimore, MD, USA. [8]Howard Hughes Medical Institute, Chevy Chase, MD, USA. [9]These authors contributed equally: Yang Liu, Oscar E. Reyes Gaido, Maximilian F. Konig. ✉e-mail: tjha@jhu.edu

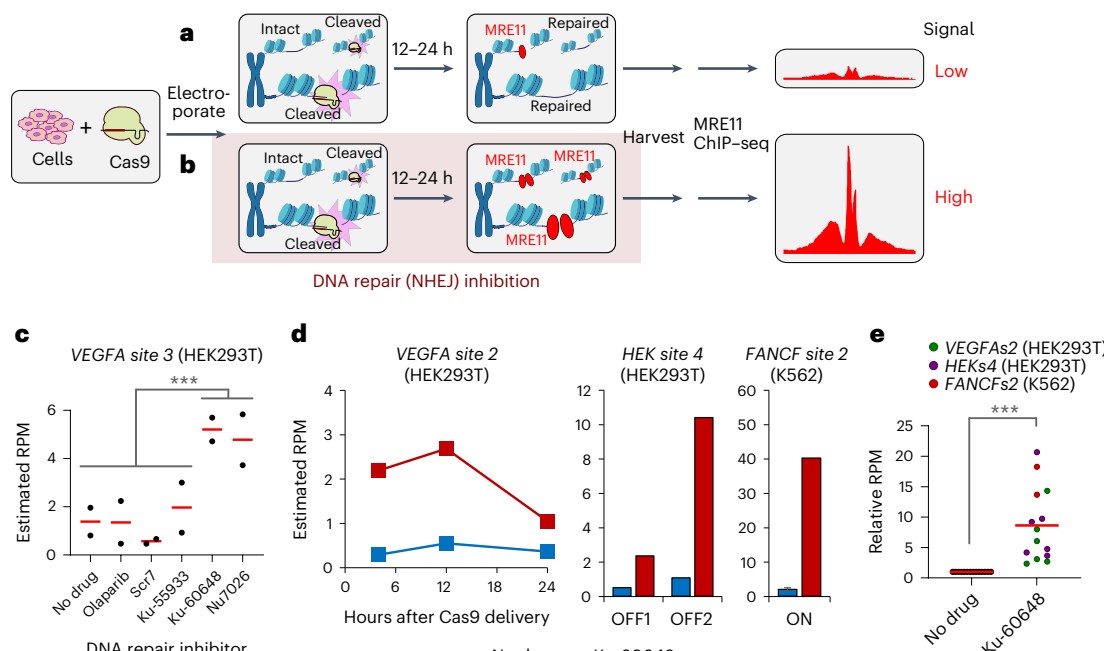

**Fig. 1 | Effect of DNA-PKcs inhibition on DNA repair at CRISPR–Cas-targeted locations. a,b,** Schematic of genome-wide CRISPR off-target detection using MRE11 ChIP–seq. **a,** Cells are unsynchronized, so only some cells have MRE11 at Cas9 cut sites at a given time (DISCOVER-Seq). **b,** Inhibition of NHEJ directs DNA repair to slower, MRE11-dependent pathways. **c,** Effect of repair factor inhibition on MRE11 residence at the *VEGFA site 3* on-target site, measured by ChIP–qPCR estimating 'reads per million' (RPM) enrichment at 12 h after Cas9 delivery in HEK293T cells. Each point corresponds to a different biologically independent replicate of a sample exposed to the DNA repair inhibitor listed in the *x* axis. Red line is the mean of two biologically independent replicates. Samples with DNA-PKcs inhibition ($n = 4$) have significantly higher estimated RPM compared with samples without DNA-PKcs inhibition ($n = 8$) using two-sided Student's *t*-test ($P = 9.65 \times 10^{-5}$). **d,** Increased MRE11 residence upon DNA-PKcs inhibition using Ku-60648 (red) versus without inhibition (blue), measured by ChIP–qPCR. Measured over multiple time points (4 h, 12 h, 24 h) after delivery of Cas9 targeting *VEGFA site 2* in HEK293T (left plot), with Cas9 targeting *FANCF site 2* in K562 at 12 h (middle plot) and with Cas9 targeting *HEK site 4* in HEK293T at 12 h (right plot). Plots display the mean over two biologically independent replicates for left and middle plots, and one biologically independent replicate for the right plot. **e,** Plot of estimated RPM enrichment normalized to the no drug sample from data in panel **d,** for sample pairs with ('Ku-60648') or without ('no drug') DNA-PKcs inhibition. Normalized RPM enrichment with DNA-PKcs inhibition was significantly higher than without inhibitor ($P = 0.0001$), using two-sided Wilcoxon signed-rank test. Red line indicates mean of $n = 14$ total samples pooled from panel **d**; green points are HEK293T, *VEGFA site 2*; purple points are HEK293T, *HEK site 4*; red points are K562, *FANCF site 2*. \*\*\*$P < 0.001$.

sensitive method to-date for CRISPR off-target detection that is directly suitable for in vivo applications[20].

## Results

### Rationale

Direct detection of the DNA repair response as a proxy for CRISPR nuclease activity has shown promise for genome-wide CRISPR off-target detection. A previous method for off-target detection, DISCOVER-Seq[14], works by detecting the genome-wide localization of MRE11, a DNA repair factor recruited to Cas9 DSB sites, using chromatin immunoprecipitation (ChIP) followed by sequencing (ChIP–seq)[17,18]. However, sensitivity is relatively low, likely because Cas9 editing is not synchronized and MRE11 resides on DNA only transiently during active repair (Fig. 1a). We hypothesized that if DNA repair could be pharmacologically modulated to encourage MRE11 residence, then MRE11 would accumulate at every Cas9-targeted site in all cells, thus enhancing detection sensitivity with ChIP–seq (Fig. 1b).

### DNA-PKcs inhibition on the CRISPR–Cas DNA damage response

To identify inhibitors of DNA repair[20] that can modulate MRE11 residence, we first delivered Cas9 with guide RNA (gRNA) targeting *VEGFA site 3* into HEK293T cells. *VEGFA site 3* and most other gRNAs used in this study were chosen because they have been well validated in earlier off-target detection methods[8–10]. We exposed cells to one of five DNA repair inhibitors, then performed ChIP with quantitative PCR (qPCR) (ChIP–qPCR) after 12 h to measure MRE11 recruitment at the target site.

Inhibition of Poly (ADP-ribose) polymerase (PARP) and ATM serine/threonine kinase (ATM) with Olaparib and Ku-55933, respectively, did not exhibit a clear effect, whereas DNA Ligase IV inhibition with Scr7 suppressed MRE11 recruitment (Fig. 1c)[21]. Notably, blocking nonhomologous end joining (NHEJ) by inhibiting DNA-dependent protein kinase catalytic subunit (DNA-PKcs) using Ku-60648 (refs. 19,22) or Nu7026 (ref. 23) significantly increased MRE11 recruitment at the target site ($P < 1 \times 10^{-4}$; two-sided Student's *t*-test) (Fig. 1c). The effect of DNA-PKcs inhibition was consistent across multiple time points (4 h, 12 h, 24 h), three other gRNAs (*VEGFA site 2, HEK site 4, FANCF site 2*) and/or another cell line (K562) ($P < 0.001$; two-sided Wilcoxon signed-rank test) (Fig. 1d,e). These results suggest that blocking NHEJ with DNA-PKcs inhibition greatly boosts MRE11 residence at Cas9-targeted sites. Among possible DNA-PKcs inhibitors, Ku-60648 was selected for subsequent experiments due to extensive literature documenting its use in diverse contexts from cell lines to mouse models[19,22].

We aimed to better characterize the effect of DNA-PKcs inhibition on repair of Cas9-mediated DNA damage. First, we used super-resolution stimulated emission depletion (STED) microscopy[24–26] to measure the localization of 53BP1 and BRCA1 foci after Cas9-induced DNA breaks in U2OS cells. 53BP1 corresponds to activation of the NHEJ pathway, whereas BRCA1 is implicated in MRE11-dependent homology-directed repair (HDR) or microhomology-mediated end joining (MMEJ)[27,28]. Using a multi-target gRNA targeting over 100 locations[29], DNA-PKcs inhibition using Ku-60648 led to a significant reduction in 53BP1 foci relative to BRCA1, consistent with suppression of NHEJ in favor of HDR/MMEJ ($P < 1 \times 10^{-4}$; two-sided Wilcoxon rank-sum test)

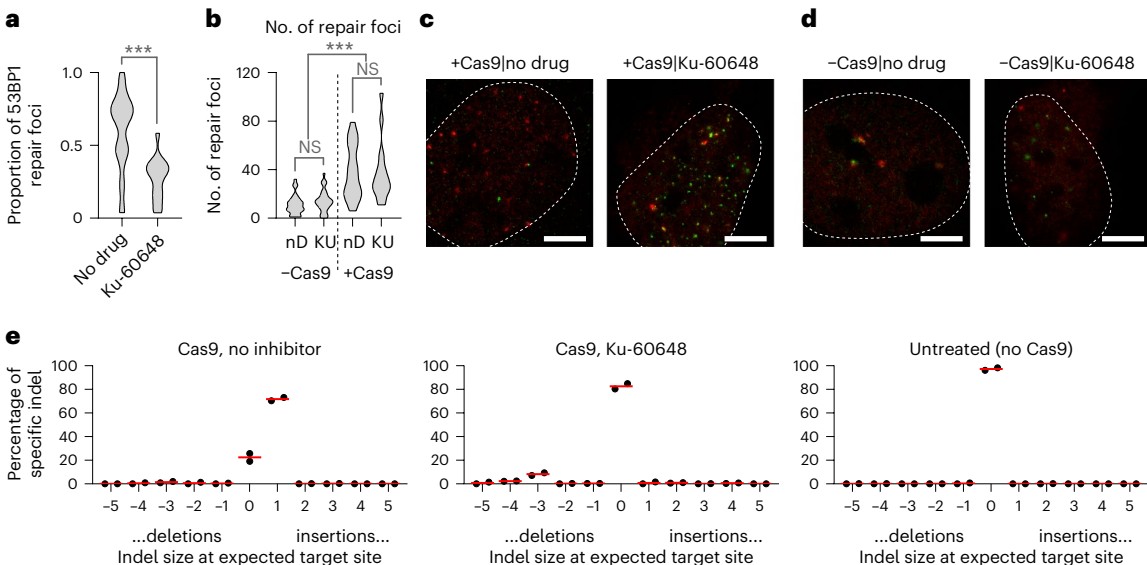

**Fig. 2 | Live cell imaging and mutation analysis with DNA-PKcs inhibition.**
**a**, The proportion of 53BP1 foci relative to BRCA1 as detected by STED in cells exposed to Cas9 targeting a multi-target gRNA with 126 genome-wide target sites. $N = 98$ cells examined over four independent experiments, $P = 0.00018$ using two-sided Wilcoxon rank-sum test. **b**, The number of repair foci (53BP1 or BRCA1) as detected by STED in cells with or without Cas9 ('+Cas9' or '−Cas9', respectively), with or without Ku-60648 ('KU' versus 'nD', respectively), targeting 126 genome-wide sites with a multi-target gRNA. $N = 98$ cells from four biologically independent replicates, $P = 4.44 \times 10^{-10}$ using two-sided Wilcoxon rank-sum test between '−Cas9' and '+Cas9'. Difference in no. of foci in each group was not significant (left, $P = 0.15$; right, $P = 0.95$). **c,d**, Representative images for panel **b**, (**c**) with Cas9 or (**d**) without Cas9. Red labels 53BP1, green labels BRCA1. Scale bar, 5 μm. **e**, Histogram of indels or no mutations ('0') at 48 h after Cas9 editing of *ACTB*, either without DNA-PKcs inhibitor ('Cas9, no inhibitor') or with inhibitor ('Cas9, Ku-60648'). Untreated cells not exposed to Cas9 shown for reference ('Untreated (no Cas9)'). Red bar displays mean over two biologically independent replicates. ***$P < 0.001$. NS, not significant.

(Fig. 2a). Ku-60648 in the absence of Cas9 did not change the number of DNA damage (53BP1 and BRCA1) foci detectable by STED, suggesting that Ku-60648 alone does not induce DNA damage inside cells (Fig. 2b–d). Additionally, we used a complementary assay to determine the type of insertion–deletion mutations (indels) by Sanger sequencing after 3 d of Cas9 targeting *ACTB* in HEK293T cells[17]. Exposure to Ku-60648 altered indel outcomes, from +1 insertions associated with NHEJ in favor of larger −3 deletions from MMEJ (Fig. 2e). Together, these results confirm that DNA-PKcs inhibition with Ku-60648 blocks the NHEJ repair pathway in favor of MRE11-associated HDR and MMEJ pathways, therefore boosting MRE11 residence.

## DNA-PKcs inhibition improves CRISPR off-target detection
Next, we determined whether increased MRE11 residence with DNA-PKcs inhibition can improve the sensitivity of CRISPR off-target discovery. At 12 h after delivery of Cas9 with *FANCF site 2* gRNA into K562 cells, we performed ChIP–seq for MRE11 followed by the BLENDER bioinformatics pipeline[14] to detect all Cas9 target sites genome-wide. Sequencing samples with or without DNA-PKcs inhibition were always normalized to the same number of reads for appropriate comparison. Treatment with Ku-60648 significantly increased MRE11 ChIP–seq enrichment at all discovered on- and off-target sites ($P < 1 \times 10^{-3}$; two-sided Wilcoxon signed-rank test), as measured by the number of reads within a 1.5-kilobase (kb) region around the target site per million total reads, that is, reads per million (RPM) (Fig. 3a–d). MRE11 ChIP–seq enrichment at a specific target site can be visualized as a histogram of base pair coverage along the genome; it exhibits two peaks on each side of the cut site because paired-end Illumina sequencing only reads the ends of DNA fragments that are enriched around the cut site (Extended Data Fig. 1a)[14]. MRE11 levels 10 kb away from the target sites did not significantly increase ($P \geq 0.18$; two-sided Wilcoxon signed-rank test), further supporting the lack of additional DNA damage caused by the inhibitor itself (Fig. 3e,f).

To reduce the likelihood of reporting false positive sites, MRE11 ChIP–seq was also performed on cells with the same experimental conditions except without Cas9 (Extended Data Fig. 1b). The final set of off-target sites is therefore determined as the set from the sample with Cas9 subtracted by the set from the corresponding sample without Cas9. For the *VEGFA site 2* gRNA, 178 sites were discovered with DNA-PKcs inhibition using Ku-60648, which is an over fivefold increase compared with 35 sites discovered without DNA-PKcs inhibition (that is, DISCOVER-Seq) (Fig. 3g and Extended Data Fig. 2a). Improved performance with Ku-60648 was consistent across different gRNAs and multiple time points (Fig. 3h,i and Extended Data Fig. 2b). The discovered sites with Ku-60648 included almost all the sites identified using DISCOVER-Seq alone (Fig. 3j). Reassuringly, only a small minority (average of 1.7%) of the initial sites were also found in corresponding negative control samples without Cas9, and therefore deemed to be false positives and removed (Extended Data Fig. 3a,b). We therefore use the term DISCOVER-Seq+ to denote CRISPR off-target discovery that combines MRE11 ChIP–seq (that is, DISCOVER-Seq)[14] with DNA-PKcs inhibition to achieve improved detection sensitivity.

Next, we assessed if any of the new sites discovered by DISCOVER-Seq+ harbor evidence of mutagenesis after CRISPR genome editing. For the *FANCF site 2* gRNA, DISCOVER-Seq+ identified 15 target sites, compared with only two with DISCOVER-Seq (Fig. 4a). We exposed cells to Cas9 targeting *FANCF site 2* for 4 d (without Ku-60648), then measured indel mutations at each discovered target site using deep amplicon sequencing[30]. Of the 13 off-target sites exclusively discovered by DISCOVER-Seq+, five exhibited detectable indels by amplicon sequencing (Fig. 4b). These results demonstrate that DISCOVER-Seq+ identified new off-target sites with evidence of indel mutations, which DISCOVER-Seq alone failed to detect. Although some newly discovered off-target sites lacked detectable indel mutations by amplicon sequencing, they are still essential to identify because DSBs, even in the absence of mutagenesis, are detrimental to the cell[21,29].

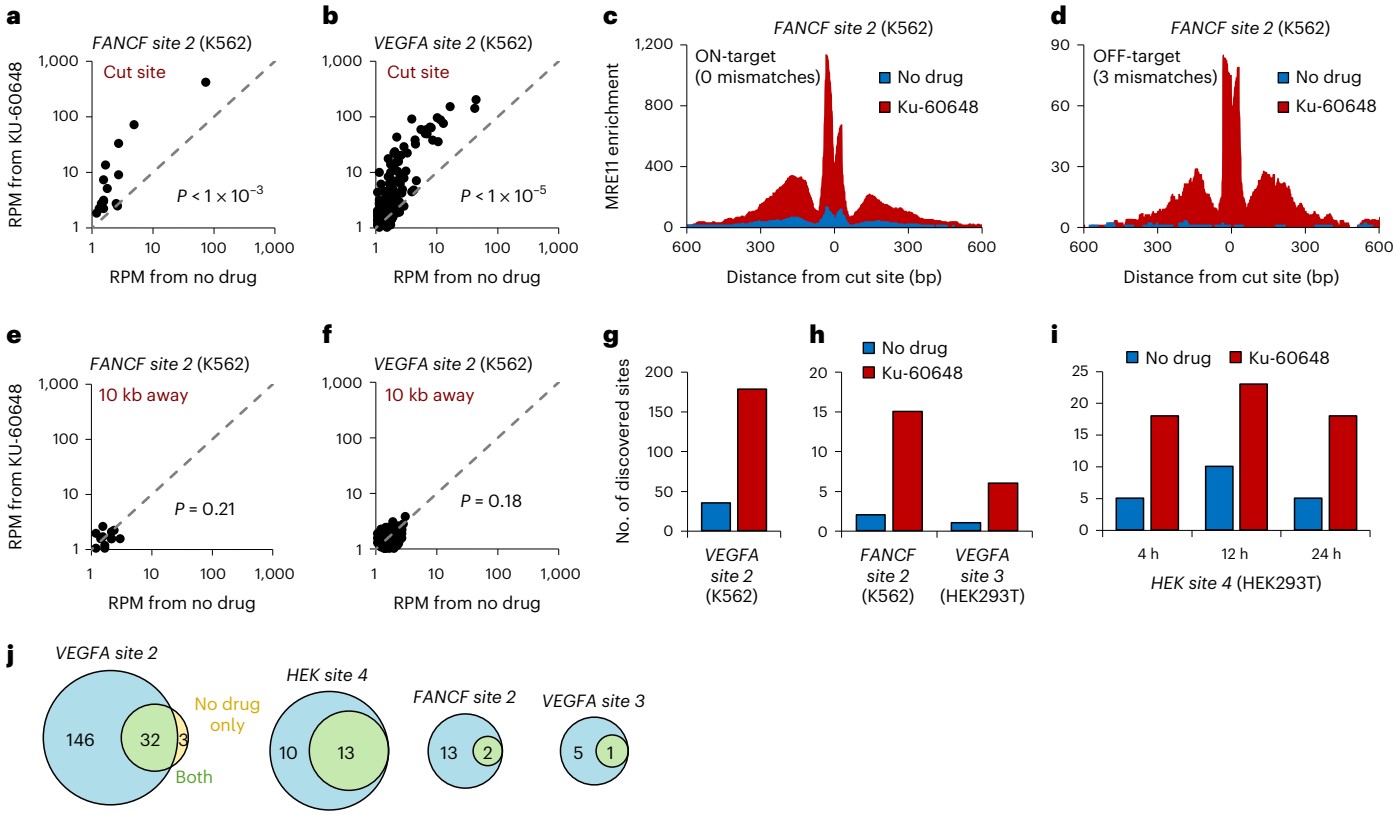

**Fig. 3 | DNA-PKcs inhibition increases the sensitivity of CRISPR off-target detection. a,b,** Plots of MRE11 ChIP−seq enrichment (number of reads within a 1.5-kb window centered at the cut site, per 1 million total reads, that is, RPM) for samples with (*y* axis) or without (*x* axis) DNA-PKcs inhibition at all *FANCF site 2* (**a**) or *VEGFA site 2* (**b**) Cas9 target sites in K562 detected from the DNA-PKcs inhibited samples. Each point in the plot (15 in panel **a**, 178 in panel **b**) corresponds to a putative target site. Significant differences ($P = 0.00081$ or $P = 4.57 \times 10^{-26}$) between *y*-axis and *x*-axis values were determined using two-sided Wilcoxon signed-rank test. **c,d,** Genome browser visualization of MRE11 enrichment at an on-target (**c**) and representative off-target (**d**) position from K562 with Cas9 targeting *FANCF site 2*, with (red) or without (blue) DNA-PKcs inhibition. **e,f,** Plots of MRE11 ChIP−seq enrichment for samples with (*y* axis) or without (*x* axis) DNA-PKcs inhibition at positions 10 kb downstream from the actual (**e**) *FANCF site 2* or

(**f**) *VEGFA site 2* cut sites, to measure background enrichment adjacent to cut sites. MRE11 enrichment with (*y* axis) versus without (*x* axis) DNA-PKcs inhibition at the adjacent background locations was not significantly different ($P = 0.21$ or $0.18$), determined using two-sided Wilcoxon signed-rank test. **g,** Number of discovered off-target sites with (red) or without (blue) DNA-PKcs inhibition for *VEGFA site 2*. Quantification of Extended Data Fig. 2a. **h,i,** Number of discovered off-target sites with (red) or without (blue) DNA-PKcs inhibition for *VEGFA site 3, FANCF site 2* (**h**) and *HEK site 4* (**i**) gRNAs. Quantification of Extended Data Fig. 2b. **j,** Venn diagram illustrating overlap in the identity of Cas9 target sites discovered from samples with DNA-PKcs inhibition ('DNA-PKi only'; light blue), without DNA-PKcs inhibition ('no drug only'; light yellow) or found in both samples ('both'; light green). Four gRNAs were evaluated.

The validity of DISCOVER-Seq+ off-target sites was further confirmed by comparing with published results by an independent technique, GUIDE-seq[8], which is notably not compatible with primary cells or in vivo applications[14]. For both the *FANCF site 2* and *VEGFA site 2* gRNAs, half or more of target sites found by DISCOVER-Seq+ were also found by GUIDE-seq, and vice versa. (Fig. 4a,c). These results demonstrate robust overlap in the discovered target sites between GUIDE-seq and DISCOVER-Seq+, providing external validity for DISCOVER-Seq+ while confirming the superiority of DISCOVER-Seq+ in identifying off-target sites.

**DISCOVER-Seq+ in editing of primary human cells**
We further evaluated the utility of DISCOVER-Seq+ in three applications: patient-derived induced pluripotent stem cell (iPSC) editing, generating engineered T cells for cancer immunotherapy and in vivo characterizations of CRISPR-based therapies in mouse models. First, we used DISCOVER-Seq+ to improve off-target detection in iPSCs. DISCOVER-Seq+ in WTC-11 iPSCs[31] discovered over twofold more off-target sites at *VEGFA site 2* compared with DISCOVER-Seq (Fig. 5a). At all discovered off-target sites, MRE11 ChIP−seq enrichment was also significantly increased ($P < 1 \times 10^{-5}$; two-sided Wilcoxon signed-rank test) (Fig. 5b,c and Extended Data Fig. 3c). For the same *VEGFA site 2*

gRNA, there were differences in off-target sites between three different cell lines (HEK293T, K562 and WTC-11 iPSC) (Extended Data Fig. 3d,e).

Next, we applied DISCOVER-Seq+ ex vivo to knock-in of a cancer neoantigen-specific transgenic T cell receptor (tgTCR) construct into primary human T cells[32–34]. We electroporated Cas9 targeting *TRA* (T Cell Receptor Alpha Locus) along with a 4,699-base pair (bp) homology-directed repair template (HDRT) encoding a tgTCR specific for HLA-A*02 loaded with mutant p53 R175H peptide[32], then performed DISCOVER-Seq+ 12 h later (Fig. 5d). The specific R175H mutation that is targeted by the tgTCR is the most prevalent p53 gain-of-function mutation in human cancers[33]. DISCOVER-Seq+ (with Ku-60648) identified 20 off-target sites genome-wide compared with four with DISCOVER-Seq (Fig. 5e), and led to significantly greater MRE11 enrichment at all discovered sites ($P = 1 \times 10^{-3}$; two-sided Wilcoxon signed-rank test) (Fig. 5f–h). In contrast, samples without Cas9 exhibited no change in enrichment with Ku-60648, further confirming that the inhibitor alone does not induce damage ($P = 0.69$; two-sided Wilcoxon signed-rank test) (Fig. 5i).

DISCOVER-Seq+ also has the potential to compare off-target profiles between different types of CRISPR nucleases. As a proof of concept, we compared the performance of Cas9 with Cas12a (Cpf1), targeting the same position in *TRA*. DISCOVER-Seq+ at 12 h after Cas12a

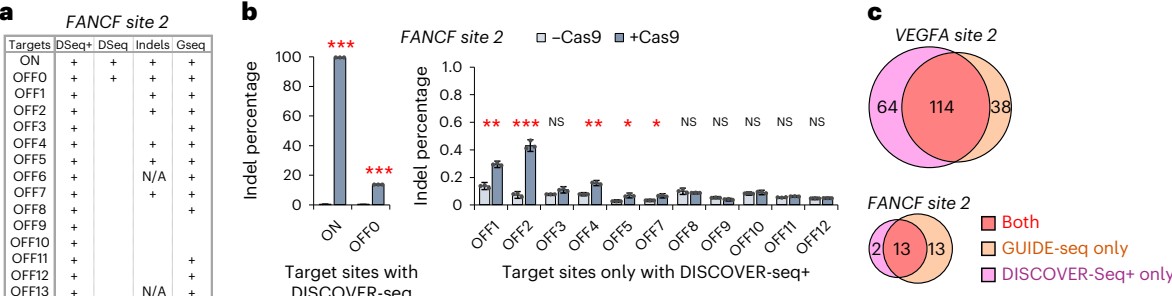

**Fig. 4 | Comparing DISCOVER-Seq+ with GUIDE-seq and amplicon sequencing. a**, For the 15 target sites (1 on-target, 14 off-targets) of the *FANCF site 2* gRNA identified by DISCOVER-Seq+ ('DSeq+'), the chart shows which sites are also identified by DISCOVER-Seq alone ('+' under 'DSeq'), which sites have indels detectable by targeted deep sequencing ('+' under 'Indels') and which sites were also detectable by GUIDE-seq ('+' under 'Gseq'). Target sites labeled with 'N/A' under 'Indels' were unable to be successfully amplified by PCR for targeted sequencing. **b**, Left plot, measurement of indels at the on-target and sole off-target site (OFF0) discovered by the original DISCOVER-Seq, for K562 cells with the *FANCF site 2* gRNA, with or without Cas9 ('+Cas9' or '−Cas9', respectively). Right plot, measurement of indels at off-target sites exclusively discovered by DISCOVER-Seq+. Plots display the mean of three biologically independent replicates; error bars represent ±1 s.d. from mean. *$P < 0.05$, **$P < 0.01$ and ***$P < 0.001$, using two-sided Student's $t$-test (exact $P$ values in the Source Data for this figure). **c**, Venn diagram illustrating overlap in the identity of *VEGFA site 2* and *FANCF site 2* target sites identified by DISCOVER-Seq+ versus GUIDE-seq.

**Fig. 5 | DISCOVER-Seq+ in human iPSCs and primary T cells. a**, *VEGFA site 2* Cas9 target sites detected using DISCOVER-Seq (left) versus DISCOVER-Seq+ (right) in WTC-11 iPSCs. **b,c**, Genome browser visualization of MRE11 enrichment at an on-target (**b**) and representative off-target (**c**) position with four mismatches ('4 mm') in WTC-11 iPSCs with Cas9 targeting *VEGFA site 2*. DISCOVER-Seq+ data in red (with Ku-60648), DISCOVER-Seq data in blue (with no drug exposure). **d**, Schematic of the DISCOVER-Seq+ protocol in the knock-in of a cancer neoantigen-specific tgTCR into the *TRA* locus of primary human T cells. **e**, *TRA* Cas9 target sites in primary T cells detected using DISCOVER-Seq (left) versus DISCOVER-Seq+ (right). **f**, Genome browser visualization of MRE11 enrichment at a representative four-mismatch ('4 mm') off-target position in primary human T cells with Cas9 targeting *TRA* for knock-in of a tgTCR template. DISCOVER-Seq+ data in red (with Ku-60648), DISCOVER-Seq data in blue (with no drug exposure). **g**, Same as panel **f**, at another four-mismatch off-target position. **h**, Plot of MRE11 ChIP–seq RPM enrichment within a 1.5-kb window for samples with ($y$ axis) or without ($x$ axis) DNA-PKcs inhibition, at all *TRA* Cas9 off-target sites in primary human T cells from the DNA-PKcs inhibited samples. Each point in the plot (20 total) corresponds to a putative target site. Differences ($P = 1 \times 10^{-3}$ or $P = 0.69$) between $y$-axis and $x$-axis values were determined using two-sided Wilcoxon signed-rank test. **i**, Same as panel **h**, for cells delivered with Cas9 but without gRNA (negative control). **j**, *TRA* Cas12a (Cpf1) target sites in primary T cells.

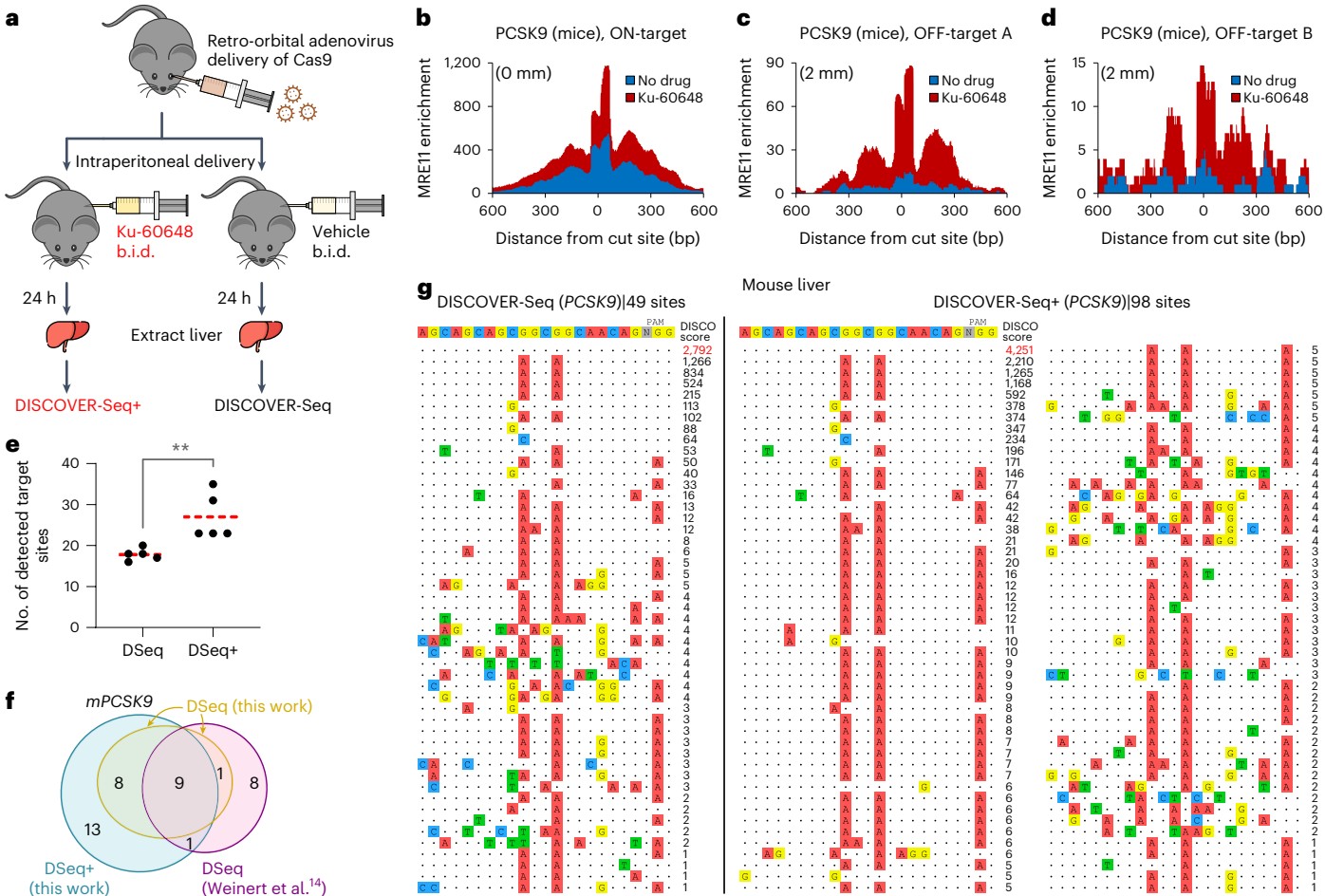

**Fig. 6 | DISCOVER-Seq+ in mice. a**, Schematic of DISCOVER-Seq+ protocol in mice. **b–d**, Genome browser visualization of MRE11 enrichment at the (**b**) *PCSK9* on-target site ('ON-target'), (**c**) one off-target site ('OFF-target A') and (**d**) another off-target site ('OFF-target B') with two mismatches each ('2 mm'), in the liver of mice transduced with adenovirus expressing Cas9 targeting *PCSK9*. Mice were dosed twice a day (b.i.d.) with either 25 mg kg⁻¹ Ku-60648 ('Ku-60648'; red) or with vehicle ('no drug'; blue). **e**, Number of detected genome-wide target sites in the mouse genome mm10 with Cas9/gRNA targeting *PCSK9*, identified using DISCOVER-Seq (DSeq) versus DISCOVER-Seq+ (DSeq+). *N* = 5 biologically independent replicates (mice) were used for each condition; two-sided Student's

*t*-test was used (*P* = 0.0079). **f**, Venn diagram illustrating overlap in the mouse *PCSK9* target sites identified by in vivo DISCOVER-Seq+ in this work, 'DSeq+ (this work)' (blue); by in vivo DISCOVER-Seq in this work, 'DSeq (this work)' (yellow); and in the original DISCOVER-Seq manuscript, 'DSeq (Wienert et al.[14])' (purple). All the target sites identified by 'DSeq (this work)' are also found in the other two groups. **g**, *PCSK9* Cas9 target sites detected using DISCOVER-Seq (left) versus DISCOVER-Seq+ (right) in mouse liver. Off-target detection for each condition was performed on sequencing data pooled across five biologically independent mouse replicates. \*\**P* < 0.01.

editing only identified the on-target site and no off-target sites (Fig. 5j), consistent with the improved specificity of Cas12a (ref. 35). Flow cytometry for tgTCR expression after 7 d in T cells without Ku-60648 exposure showed similar tgTCR integration efficiencies of 8.4% for Cas12a and 9.7% for Cas9 (Extended Data Fig. 3f–h). Together, our preliminary analysis using DISCOVER-Seq+ revealed that Cas12a maintained adequate tgTCR integration rates while eliminating detectable off-target damage. Importantly, these experiments demonstrated that DISCOVER-Seq+ is directly compatible with CRISPR knock-in using a homology template in primary human T cells.

### DISCOVER-Seq+ in vivo

Finally, we evaluated DISCOVER-Seq+ in vivo by targeting the cardiovascular risk gene *PCSK9* in mouse liver[36]. We retro-orbitally injected adenovirus encoding Cas9 and *PCSK9* gRNA into ten, 8–10-week-old, male C57BL/6J mice, followed by peritoneal injection of either 25 mg kg⁻¹ Ku-60648 (that is, DISCOVER-Seq+) or vehicle (that is, DISCOVER-Seq) twice daily (b.i.d.) (Fig. 6a). We selected the specific *PCSK9* gRNA that was also used in the original DISCOVER-Seq study for direct comparison[13]. Ku-60648 has been evaluated as a drug for chemo-sensitization in

cancer therapy, exhibits good pharmacokinetics and strongly penetrates tissue including tumors[22]. Mice were killed after 24 h to collect the liver for MRE11 ChIP–seq. DISCOVER-Seq+ mice exhibited increased MRE11 ChIP–seq signal in their liver compared with those without DNA-PKcs inhibition (*P* < 1 × 10⁻⁴; two-sided Wilcoxon rank-sum test) (Fig. 6b–d and Extended Data Fig. 3i). An average of 27 target sites were identified with DISCOVER-Seq+ compared with 18 sites with DISCOVER-Seq across five biologically independent replicates (*P* < 0.01; two-sided Student's *t*-test) (Fig. 6e). The identified sites strongly overlap between the two methodologies and with sites identified in the original DISCOVER-Seq study (Fig. 6f) (ref. 14). Pooling sequencing reads across all five replicates identified 98 target sites with DISCOVER-Seq+ versus 49 with DISCOVER-Seq (Fig. 6g). Together, these results demonstrate that DISCOVER-Seq+ is compatible with direct measurement of genome-wide off-target editing in vivo (Supplementary Table 1).

### Discussion

This study designed and validated DISCOVER-Seq+, the most sensitive method to-date for detecting CRISPR–Cas off-target activity in primary cells and in vivo, to our knowledge. As CRISPR becomes an increasingly

feasible approach for therapeutic genome editing[1,20,32,33,36–38], evaluation of CRISPR off-target activity directly in clinically translatable applications is crucial. Even if a specific CRISPR gRNA does not have detectable off-target activity in one particular cell type, this may not translate to other cell types, tissues or organisms. Directly measuring off-target sites in the system of interest, whether in primary T cells, mice or even nonhuman primates, is therefore essential. By combining unparalleled detection sensitivity with high versatility in ex vivo and in vivo applications, we believe DISCOVER-Seq+ will find widespread use as the state-of-the-art technology for off-target detection in diverse applications of CRISPR editing.

DISCOVER-Seq+ identified off-target sites with evidence of mutagenesis that DISCOVER-Seq alone failed to detect[8,39]. Furthermore, because DISCOVER-Seq+ directly measures off-target DNA damage rather than mutagenesis, it also discovered sites that did not have detectable indel mutations. This is important for three reasons: (1) One study found that less than 15% of DSBs convert to indels in a single damage cycle[17]; therefore, off-target sites that are not frequently damaged may not result in indels that can appear in amplicon sequencing-based queries. (2) DNA damage outcomes such as DSBs are highly detrimental regardless of mutagenesis, especially to primary cells, by inducing widespread epigenetic changes and perturbing native cellular functions such as cell division and transcription[21,29]. (3) Evaluating indels alone may also miss complex DNA damage outcomes such as large deletions and translocations[8,40]. Furthermore, deep amplicon sequencing may miss rarer off-target sites due to a lack of enrichment of altered DNA molecules and insufficient sequencing depth[41]. For all these reasons, enrichment for sites with evidence of CRISPR–Cas-induced DNA damage is a more holistic readout of off-target activity than indels alone. In summary, DISCOVER-Seq+ directly detects genome-wide CRISPR-induced DNA damage, regardless of whether these sites become mutated, with unprecedented sensitivity and versatility.

DNA-PKcs inhibition could also influence the performance of other methods to detect CRISPR–Cas activity. BLISS/BLESS[9] could be improved with DNA-PKcs inhibition by increasing the quantity of unrepaired DSBs at the time of evaluation. In contrast, GUIDE-seq[8] would likely be impaired because incorporation of its double-stranded oligodeoxynucleotide relies on NHEJ, which is directly inhibited by DNA-PKcs inhibitors. Future studies should explore whether other strategies of inhibiting DNA repair may improve detection of CRISPR–Cas off-target activity.

Limitations of DISCOVER-Seq+ include the need for the DNA-PKcs inhibitor to exert its effect, including in vivo. We believe this is not a major concern because Ku-60648 is a small molecule previously shown to have good bioavailability and tissue penetration, including into tumors[22,23]. In contrast, the main barrier to applicability in other organs in vivo remains the efficiency of CRISPR–Cas delivery to nonliver organs[1,14,36]. In addition, the improvement in the number of discovered off-target sites with DNA-PKcs inhibitor in mice[14] is a factor of two, which is lower than in cell lines. This may be due to reduced inhibitor or Cas9 concentrations in the liver; further optimization of drug dosing and Cas9 delivery may improve performance.

Identifying off-target genome editing is a major barrier to applications of CRISPR–Cas systems. By leveraging MRE11 ChIP–seq with DNA-PKcs inhibition, DISCOVER-Seq+ provides the highest detection sensitivity to-date in systems ranging from ex vivo editing of primary human cells to in vivo editing of mice, setting the standard for genome-wide CRISPR off-target discovery. DISCOVER-Seq+ has the potential to validate the specificity profile of genome editing at numerous stages of the therapeutic development pipeline, from cell lines and primary cells to mice and potentially nonhuman primates[1,18,32,33,36–38].

## Online content

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

## Methods

### Cell culture

HEK293T cells (ATCC CRL-3216) and K562 cells (ATCC CCL-243) were cultured at 37 °C under 5% $CO_2$ in DMEM (Corning) supplemented with 10% FBS (Clontech), 100 units per ml of penicillin and 100 µg ml$^{-1}$ streptomycin (DMEM complete). Cells were tested every month for mycoplasma.

A human iPSC (hiPSC), the WTC-11 cell line (GM25256, Coriell Institute) (Kreitzer et al., 2013), was used for all iPSC experiments in this study. We followed the guidelines of Johns Hopkins Medical Institute for the use of this hiPSC line. Briefly, frozen WTC-11 cells were first thawed in a 37 °C water bath and washed in Essential 8 Medium (E8 medium; Thermo Fisher Scientific, no. A1517001) by centrifugation. After resuspension, WTC-11 cells were plated onto a 6-cm cell culture dish pre-coated with human embryonic cell-qualified Matrigel (1:100 dilution, Corning, no. 354277). Plate coating should be performed for at least 2 h. Subsequently, 10 µM ROCK inhibitor (Y-27632; STEMCELL, no. 72308) was supplemented into the E8 medium to promote cell growth and survival. For subculture, WTC-11 cells were dissociated from the plate using Accutase (Sigma, no. A6964) and passaged every 2 d. WTC-11 cells were maintained in an incubator at 37 °C with 5% $CO_2$.

### Ethics statement and mouse husbandry

All mouse studies were carried out in accordance with guidelines and approval of the Johns Hopkins University Animal Care and Use Committee (Protocol no. MO20M274). The 8–10-week-old male C57BL/6J mice (The Jackson Laboratory) were housed in a facility with 12-h light/12-h dark cycle at 22 ± 1 °C and 40 ± 10% humidity. Teklad Global 18% protein rodent diet and tap water were provided ad libitum.

### Immunofluorescence and imaging by STED microscopy

U2OS cells stably expressing Cas9-EGFP cells were seeded onto 35-mm, glass bottom dishes and transfected with multi-target guide RNAs (mtgRNAs) for 12–24 h. Cleavage was activated by ultraviolet light for 1 min. To fix cells, 4% pre-warmed paraformaldehyde in 1 × PBS was used for 10 min. After rinsing three times with 1 × PBS, cell membrane permeabilization was performed with Triton-X used for 10 min. Then, 2% w/v BSA in 1 × PBS was used for blocking for 1 h and at room temperature. The primary antibodies, mouse anti-BRCA1 (sc-6954 D9, Santa Cruz Biotechnology), rabbit anti-53BP1 (ab172580, Abcam), were diluted (1:500) in 1 × PBS and directly added into the imaging dish. After 1 h of incubation, the primary antibody was removed, and the sample was washed with 1 × PBS three times. The samples were then incubated for 30 min with the secondary antibodies, goat anti-Mouse Alexa-594 (A-21235, ThermoFisher) and goat anti-Rabbit Atto-647N (40839, Sigma), diluted (1:1,000) in 1 × PBS. Finally, the sample was rinsed three times and mounted with Prolong Diamond mounting medium (Thermo Fisher Scientific) overnight.

All STED images were obtained using a home-built two-color STED microscope (Han and Ha[24]; Ma and Ha[25]). In short, a femtosecond laser beam with a repetition rate of 80 MHz from a Ti:Sapphire laser head (Mai Tai HP, Spectra-Physics) is split into two parts: one part produces an excitation beam coupled into a photonic crystal fiber (Newport) for wide-spectrum light generation. The beam is further filtered by a frequency-tunable acoustic optical filter (AA Opto-Electronic) for multi-color excitation. The other part of the laser pulse is temporally stretched to ~300 ps (with two 15-cm-long glass rods and a 100-m-long polarization-maintaining single-mode fiber, OZ optics), collimated and expanded, and wave-front modulated with a vortex phase plate (VPP-1, RPC photonics). This modulation produces a hollow STED spot generation to de-excite the fluorophores at the periphery of the excitation focus, thus improving the lateral resolution. The STED beam is set at 765 nm with a power of 120 mW at the back focal plane of the objective (NA = 1.4 HCX PL APO 100X, Leica), and the excitation wavelengths are set as 594 nm

and 650 nm for imaging Alexa-594- and Atto-647N-labeled targets, respectively. Two avalanche photodiodes detect the fluorescent photons (SPCM-AQR-14-FC, Perkin Elmer). The images are obtained by scanning a piezo-controlled stage (Max311D, Thorlabs) controlled with the Imspector data acquisition program.

### Electroporation of Cas9 RNP and DNA-PKcs inhibitor delivery into cell lines and iPSCs

CRISPR RNA (crRNA) and trans-activating crRNA (tracrRNA) sequences are listed in Supplementary Table 2. First, 2 µl of 100 µM crRNA was mixed with 2 µl of 100 µM tracrRNA (Integrated DNA Technologies) and heated to 95 °C for 5 min in a thermocycler, then allowed to cool on the benchtop for 5 min. To form the ribonucleoprotein (RNP) complex, 3 µl of 10 µg µl$^{-1}$ (~66 µM) purified Cas9 was mixed with the annealed 4 µl of 50 µM cr:tracrRNA, then 8 µl of dialysis buffer (20 mM HEPES pH 7.5 and 500 mM KCl, 20% glycerol) was mixed in for a total of 15 µl. This solution was incubated for 20 min at room temperature to allow for RNP formation.

HEK293T cells were maintained to a confluency of ~90% before electroporation. Twelve million cells were trypsinized with 5 min of incubation in the incubator, then 1:1 of DMEM complete was added to inactivate trypsin. This mixture was centrifuged (3 min, 200g) and supernatant removed, followed by resuspension of the cell pellet in 1 ml of PBS, centrifugation (3 min, 200g) and finally complete removal of supernatant. Then, 90 µl of nucleofection solution (16.2 µl of Supplement solution mixed with 73.8 µl of SF solution from SF Cell Line 4D-Nucleofector X Kit L, Lonza) was mixed thoroughly with the cell pellet. The 15 µl of RNP solution was mixed in along with 2 µl of Cas9 Electroporation Enhancer (Integrated DNA Technologies). The entirety of the final solution (approximately 125 µl) was transferred to one well of a provided cuvette rated for 100 µl. Electroporation was then performed according to the manufacturer's instructions on the 4D-Nucleofector Core Unit (Lonza) using code CA-189. Some white residue may appear in the cell mixture after electroporation, but that is completely normal. A total of 400 µl of DMEM complete was used to completely transfer the cells out of the cuvette, before plating to culture wells pre-coated with 1:100 collagen. A minimum of 4 million cells are used for each ChIP. For time-resolved experiments, this means one electroporation equates to three samples.

For WTC-11 iPSCs, cells were dissociated from the plate using accutase (Sigma, no. A6964). Electroporation was performed using the Lonza P3 Primary Cell 4D-Nucleofector X Kit L using code CA-137, on 10 million cells, and using 65 µl of the P3 solution mixture with EP enhancer per electroporation cuvette (compared with 90 µl of comparable SF solution mixture for HEK293T cells). After electroporation, cells were resuspended in E8 medium supplemented with 10 µM ROCK inhibitor (Y-27632; STEMCELL, no. 72308), and plated onto a 10-cm cell culture dish pre-coated with human embryonic cell-qualified Matrigel (1:100 dilution, Corning, no. 354277) for at least 2 h.

To expose cells to DNA repair inhibitors, they were added to the culture media at a final concentration of 1 µM KU-60648 (1:2,500 of 2.5 mM KU-60648), 20 µM Nu7026 (1:500 of 10 mM Nu7026), 10 µM Ku-55933 (1:10,000 of 100 mM KU-55933), 1 µM Scr7 (1:10,000 of 10 mM Scr7) or 10 µM Olaparib (1:1,000 of 10 mM Olaparib). All stock solutions of drug were diluted in dimethylsulfoxide.

### Adenovirus and DNA-PKcs inhibitor delivery into mice

For in vivo gene delivery, 8–10-week-old mice were anesthetized with 2.5% isofluorane/oxygen mixture. Mice received a single retro-orbital injection of $1 \times 10^9$ infectious adenoviral particles (Ad-Cas9-U6-mPCSK9-sgRNA) in 100 µl of sterile saline. Immediately following, mice received intraperitoneal delivery of KU-60648 dosed at 25 mg kg$^{-1}$ (or vehicle only) in 100 µl of citrate buffer, or 100 µl of citrate buffer vehicle. Mice received a dose of KU-60648 or vehicle every 12 h via intraperitoneal injection.

## Extraction of mouse liver into cell suspension

At the experimental endpoint of 12 h, mice were anesthetized with isofluorane and euthanized via cervical dislocation. Liver tissue was collected, washed three times in 2 ml of PBS with 1 × protease inhibitor (Halt Protease Inhibitor Cocktail, Thermo), then the tissue disrupted in 1 ml of PBS with 1 × protease inhibitor using a loose-fitting Dounce homogenizer. For MRE11 ChIP–seq, homogenized tissue was placed on ice and used immediately.

## DISCOVER-Seq+, DISCOVER-Seq and MRE11 ChIP–seq

The protocol was adapted from previous literature (Wienert et al.[14]) and describes the reagents for one MRE11 ChIP–seq experiment. No animals or data points were excluded from the analysis.

For adherent cells, approximately 10 million cells were gently rinsed with room temperature PBS, washed off the plate using 10 ml of DMEM with assistance from pipette squirts and a cell scraper, then transferred to a 15-ml Falcon tube. For suspension cells, approximately 10 million cells were transferred to a 15-ml Falcon tube, spun down at 200$g$ for 1 min, decanted, then resuspended with 10 ml of DMEM. Then, 721 µl of 16% formaldehyde (methanol-free) was added and the tube was mixed by inversion at room temperature: 7 min for WTC-11 iPSCs, 12 min for HEK293T cells or 15 min for K562 cells. For mouse liver, 300 µl of Dounce-homogenized mouse liver was diluted into 10 ml of PBS. Then, 721 µl of 16% formaldehyde (methanol-free) was added and mixed by inversion at room temperature for 10 min.

Afterwards, 750 µl of 2 M glycine was added to quench the formaldehyde. Cells were spun down at 1,200$g$ and 4 °C for 3 min, then washed with ice-cold PBS twice, spinning down with the same centrifugation conditions. Pellets can be decanted, flash-frozen, then stored at −80 °C for later use. Cells were then resuspended in 4 ml of lysis buffer LB1 (50 mM HEPES, 140 mM NaCl, 1 mM EDTA, 10% glycerol, 0.5% Igepal CA-630, 0.25% Triton X-100, pH to 7.5 using KOH, then add 1 × protease inhibitor right before use) for 10 min at 4 °C, then spun down at 2,000$g$ and 4 °C for 3 min. The supernatant was decanted. Cells were then resuspended in 4 ml of LB2 (10 mM Tris-HCl pH 8, 200 mM NaCl, 1 mM EDTA, 0.5 mM EGTA, pH to 8.0 using HCl, then add 1 × protease inhibitor right before use) for 5 min at 4 °C, spun down with the same protocol and the supernatant decanted. Cells were then resuspended in 1.5 ml of LB3 (10 mM Tris-HCl pH 8, 100 mM NaCl, 1 mM EDTA, 0.5 mM EGTA, 0.1% Na-deoxycholate, 0.5% N-lauroylsarcosine, pH to 8.0 using HCl, then add 1 × protease inhibitor right before use) and transferred to 2-ml Eppendorf tubes for sonication with 50% amplitude, 30 s ON, 30 s OFF for 12 min total time (Fisher 150E Sonic Dismembrator). Sample was spun down at 20,000$g$ and 4 °C for 10 min, and supernatant was transferred to 1.5 ml of LB3 in a 15-ml falcon tube. Then, 300 µl of 10% Triton X-100 was added, and the entire solution was well mixed by gentle inversion.

Beads pre-loaded with antibodies were prepared before cell collection. First, 50 µl of Protein A beads (ThermoFisher) were used per immunoprecipitation and transferred to a 2-ml Eppendorf tube on a magnetic stand. Beads were washed twice with blocking buffer BB (0.5% BSA in PBS), then resuspended in 100 µl of BB per immunoprecipitation. Then, 4 µl of MRE11 antibody (Novus NB100-142) per immunoprecipitation was added and placed on a rotator for 1–2 h. Right before immunoprecipitation, the 2-ml tube was placed on a magnetic rack and washed three times with BB, before resuspending in 50 µl of BB per electroporation (EP). Next, 50 µl of beads in BB were transferred to each 3-ml immunoprecipitation (effective 1:750 dilution) and placed at 4 °C on a rotator for 6+ hours.

Samples were transferred to 2-ml Eppendorf tubes on a magnetic stand, washed six times with 1 ml of RIPA buffer (50 mM HEPES, 500 mM LiCl, 1 mM EDTA, 1% Igepal CA-630, 0.7% Na-deoxycholate, pH to 7.5 using KOH), then washed once with 1 ml of TBE buffer (20 mM Tris-HCl pH 7.5, 150 mM NaCl) before decanting. Beads containing DNA from ChIP were mixed with 70 µl of elution buffer EB (50 mM Tris-HCl pH

8.0, 10 mM EDTA, 1% SDS) and incubated at 65 °C for 6+ hours. Then, 40 µl of TE buffer was mixed in to dilute the SDS, followed by 2 µl of 20 mg ml$^{-1}$ RNaseA (New England BioLabs) for 30 min at 37 °C. Next, 4 µl of 20 mg ml$^{-1}$ Proteinase K (New England BioLabs) was added and incubated for 1 h at 55 °C. The genomic DNA was column purified (Qiagen) and eluted in 35 µl of nuclease-free water.

Oligo sequences for library preparation are listed in Supplementary Table 4. End-repair/A-tailing was performed on 17 µl of DNA from ChIP using NEBNext Ultra II End Repair/dA-Tailing Module (New England BioLabs), followed by ligation (MNase_F/MNase_R) with T4 DNA Ligase (New England BioLabs). Thirteen cycles of PCR using PE_i5 and PE_i7XX primer pairs were performed for MRE11 ChIP samples to amplify sequencing libraries. Samples were pooled and quantified with QuBit (Thermo), Bioanalyzer (Agilent) and qPCR (BioRad).

Cell line samples were sequenced on a NextSeq 500 (Illumina) using paired 2 × 36-bp reads. Mouse liver samples were sequenced on a DNBSEQ PE100 (BGI) using paired 2 × 50-bp reads. All ChIP–seq raw reads in FASTQ format and processed alignments in BAM format are uploaded to the Sequence Read Archive under BioProject accession PRJNA801688.

Reads were demultiplexed after sequencing using bcl2fastq. Paired-end reads were aligned to hg38, hg19 or mm10 using bowtie2. To ensure fair comparison between DISCOVER-Seq+ (with DNA-PKcs inhibitor) and DISCOVER-Seq (without inhibitor), equal numbers of sequencing reads were obtained by subsetting for each set of samples. Samtools was used to filter for mapping quality ≥ 25, remove singleton reads, convert to BAM format, remove potential PCR duplicates and index BAM-formatted output files. The software that coordinates these steps as well as performs subsequent analyses is open source (https://github.com/rogerzou/DSeqPlus).

BLENDER (Wienert et al.[14]) (https://github.com/staciawyman/blender) was used to determine Cas9 off-target sites, outputting a curated list of all off-target sites with corresponding visualization. A more sensitive cutoff threshold of 2 (-c 2) was used for all samples, except a threshold of 3 (-c 3) for the merged *PCSK9* samples.

## CRISPR–Cas9 or Cas12a editing of primary human T cells

Engineered T cells expressing a TP53 R175H:HLA-A*02:01-specific T cell receptor (TCR) under control of an EF1-alpha promoter were generated via CRISPR–Cas-mediated HDR electroporation as follows. Nucleotide sequences of the TCR of interest, promoter and homology arms for the *TRAC* gene locus were generated by de novo gene synthesis (GeneArt). A 4,699-bp HDRT double-stranded DNA was generated by amplification from a plasmid template using the Q5 High-Fidelity 2X Master Mix (New England BioLabs) with primers containing truncated Cas9 target sequences (IDT) (PMID: 31819258). Amplicon DNA was purified with AMPure beads (Beckman Coulter), eluted in water and quantified. Purified PCR products were analyzed by agarose gel electrophoresis to assess correct amplicon size and purity. T cells were isolated by negative selection using immunomagnetic cell separation (EasySep Human T Cell Isolation Kit) from cryopreserved healthy donor peripheral blood mononuclear cells collected via leukapheresis. Purified CD3$^+$ T cells were activated with Dynabeads Human T-Activator CD3/CD28 (ThermoFisher) at a 1:2 bead-to-cell ratio in RPMI-1640 (ATCC) supplemented with 10% FBS (HyClone Defined), 100 units per ml of Penicillin (Gibco), 100 µg ml$^{-1}$ Streptomycin (Gibco), 100 IU ml$^{-1}$ recombinant human IL-2 (Proleukin, Prometheus Laboratories) and 5 ng ml$^{-1}$ recombinant human IL-7 (BioLegend) at 37 °C, 5% CO$_2$. After 48 h and before electroporation, beads were removed with a magnet. Cas9 RNP targeting *TRAC* (AGAGTCTCTCAGCTGGTACA) or Cpf1 (Cas12a) RNP targeting a juxtaposed nucleotide sequence in *TRAC* (GAGTCTCTCAGCTGG-TACAC) was assembled by mixing the appropriate sgRNA (IDT) with either Alt-R S.p. Cas9 nuclease V3 (IDT) or Alt-R A.s. Cas12a (Cpf1) Ultra nuclease (IDT) and matching ssDNA Electroporation Enhancer (IDT) and incubating the mixture at room temperature for 15 min. RNPs were

mixed with 0.5 µg of the same HDRT and incubated for 5 min at room temperature. To edit activated T cells, 20 µl of T cells was resuspended in P3 buffer at $5 \times 10^7$ cells per ml (Lonza) and added to the electroporation mixture. Electroporation was performed with a 4D-Nucleofector X Unit (Lonza) in 16-well cuvettes using pulse code EH115. After electroporation, T cells were recovered by immediately adding 80 µl of warm, cytokine-free T cell medium to the cuvettes and incubation at 37 °C for 15 min. Then, T cells were diluted in T cell growth medium containing 100 IU ml$^{-1}$ recombinant human IL-2 and 5 ng ml$^{-1}$ recombinant human IL-7 in the presence of 1 µM Ku-60648 or vehicle (dimethylsulfoxide) and incubated for 12 h at 37 °C, 5% $CO_2$. A fraction of electroporated T cells for each condition was maintained in T cell growth medium with human IL-2 and IL-7 for 7 d before analysis for gene editing rates and surface expression of TP53 R175H:HLA-A*02:01-specific TCR by flow cytometry.

### High-throughput sequencing of genomic DNA samples
Genomic DNA was extracted using the DNeasy Blood & Tissue Kit (Qiagen, 69504) following manufacturer instructions. Approximately 1 million cells were used from cell lines and iPSCs. Approximately 10–20 µl of mouse liver cell suspension was used out of 1.5 ml total, and the genome extraction protocol included the Buffer ATL step for tissue lysis.

Genomic DNA samples were amplified with PCR using Q5 Hot-Start High-Fidelity 2X Master Mix (New England BioLabs, M0494). Primer pairs for all sequences are listed in Supplementary Table 3. For example, the primer set for amplifying around the *FANCF site 2* on-target site is NGS_Fs2_ON_F and NGS_Fs2_ON_R. After amplicon PCR, cleanup was performed using 1.4 × AMPure XP (Beckman Coulter A63881 https://www.ncbi.nlm.nih.gov/nuccore/A63881) following the manufacturer's instructions. Dual-indexing PCR was performed using KAPA HiFi HotStart ReadyMix (Roche, 07958935001) and PCR cleanup was performed using 1 × AMPure XP. Samples were quantified using QuBit (Thermo Fisher Scientific), pooled, diluted and loaded onto a MiSeq (Illumina). Sequencing was performed with the following number of cycles, '151|8|8|151', with the paired-end Nextera sequencing protocol.

Sequencing reads were either demultiplexed automatically using MiSeq Reporter (Illumina) or with a custom Python script to individual FASTQ files. For indel calling, sequencing reads were scanned for exact matches to two 20-bp sequences that flank +/−20 bp from the ends of the target sequence. If no exact matches were found, the read was excluded from analysis. After additional filtering for an average quality score > 20, an indel is defined as a sequence that differs in length from the reference length.

### Flow cytometry and analysis of primary human T cells
Surface staining for flow cytometry was performed by washing T cells in PBS, pH 7.4, followed by staining with LIVE/DEAD Fixable Violet Stain (ThermoFisher) at recommended concentration for 30 min on ice in the dark. T cells were then resuspended in Cell Staining Buffer (BioLegend) plus relevant antibodies (APC anti-NGFR (BioLegend)) and PE-conjugated HLA-A*02:p53 R175H tetramer (NIH Tetramer Facility) for 30 min at room temperature in the dark. Cells were washed twice in Cell Staining Buffer before resuspension for analysis. Flow cytometric analysis was performed on an IntelliCyt iQue Screener PLUS VBR (Sartorius). FlowJo v.10 was used for flow cytometry data analysis.

The gating strategy used for primary human T cells is shown in Extended Data Fig. 2. For all experiments, debris was first excluded by a morphology gate based on FSC-H and SSC-H. Nonsinglets were excluded from analysis by a single-cell gate based on FSC-H and FSC-A. Live cells were selected by gating on cells negative for staining with LIVE/DEAD Fixable Violet dye (405-nm excitation, Invitrogen). Anti-NGFR APC$^+$/tetramer PE$^+$ cells were gated to define successfully edited live, single T cells using appropriate compensation with single-stained controls.

### Statistical analyses
Student's *t*-test was used to compare samples whose distributions approximate the normal distribution, with the unpaired *t*-test for independent samples and paired *t*-test for paired samples. Wilcoxon rank-sum and signed-rank tests were used to compare independent and paired samples, respectively, whose distributions do not approximate the normal distribution. Two-sided tests were used in all cases. The *P* values for each statistical test are presented in the corresponding figure legend.

### Reporting summary
Further information on research design is available in the Nature Portfolio Reporting Summary linked to this article.

### Data availability
All sequencing data are uploaded to the Sequence Read Archive under BioProject accession PRJNA801688. All other data are available within the paper and its Supplementary Information files. Source data are provided with this paper.

### Code availability
Complete analysis code is available on GitHub (https://github.com/rogerzou/DSeqPlus).

### Acknowledgements
We thank members of the T.H., S. Myong and M. E. Anderson laboratories for support. In particular, we thank Y. Ma for assisting with STED microscopy, and J. Granger and E. Luczak for assisting with mice experiments. This work was supported by grants from the National Institutes of Health (grant nos. R35 GM 122569 and U01 DK 127432 to T.H.; grant nos. T32 GM 136577 and F30 CA 254160 to R.S.Z.; grant no. T32 GM 136577 to B.J.M.; grant no. R35 HL 140034 to M. E. Anderson) and the National Science Foundation (grant nos. PHY 1430124 and EFMA 1933303 to T.H.). Y.L. is supported by the Research Corporation for Science Advance (RCSA) as a Cottrell Postdoctoral Fellow. M.F.K. is supported by the Jerome Greene Foundation and the Cupid Foundation. A.M.-G. is a Howard Hughes Medical Institute Awardee of the Life Sciences Research Foundation. T.H. is an investigator of the Howard Hughes Medical Institute.

### Author contributions
R.S.Z. and T.H. conceived the study and oversaw experimental design. Y.L. and R.S.Z. identified the potent inhibitors and validated DISCOVER-Seq+ in immortalized cell lines and iPSCs. O.E.R.G. and R.S.Z. validated DISCOVER-Seq+ in mice. M.F.K., R.S.Z. and B.J.M. validated DISCOVER-Seq+ in primary T cells. R.S.Z., Y.L. and A.M.-G. performed ChIP–seq. Y.L. performed immunofluorescence. L.L.S. performed deep amplicon sequencing. F.A.-V. performed stimulated emission depletion (STED) microscopy. R.S.Z. performed bioinformatics and data analysis, and wrote the manuscript with contributions from all authors. T.H. supervised the study.

### Competing interests
Johns Hopkins University has submitted a patent application (PCT/US2022/079887) on the method for off-target detection used in this study with R.S.Z., Y.L. and T.H. as authors. The remaining authors declare no competing interests.

### Additional information
**Extended data** is available for this paper at https://doi.org/10.1038/s41592-023-01840-z.

# Article

**Correspondence and requests for materials** should be addressed to Taekjip Ha.

Peer reviewer reports are available. Primary Handling Editors: Lei Tang and Madhura Mukhopadhyay, in collaboration with the *Nature Methods* team.

**a**

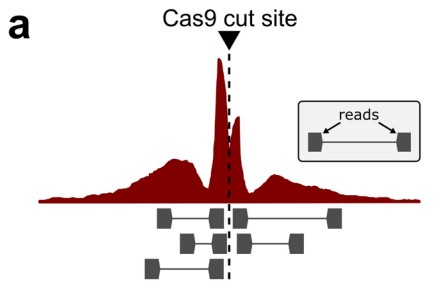

**b**

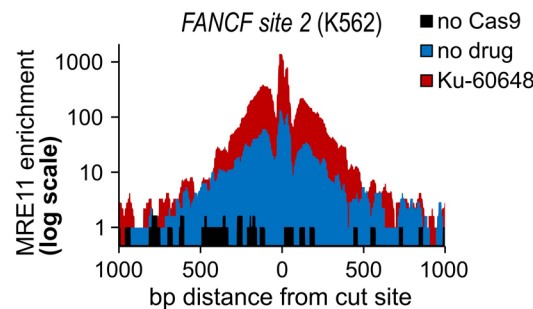

**Extended Data Fig. 1 | Illustration of MRE11 ChIP-seq enrichment. a**, Illustration of MRE11 ChIP-seq enrichment at a specific target site, visualized as a histogram of base pair coverage from sequencing reads along the genome. Only the two ends of each DNA fragment are sequenced. **b**, Genome browser visualization of MRE11 enrichment at the on-target site in K562 cells with Cas9 targeting *FANCF site 2*. Y-axis is in $\log_{10}$ scale. Overlaying red and blue graphs correspond to with or without DNA-PKcs, respectively. Black graph corresponds to samples without Cas9.

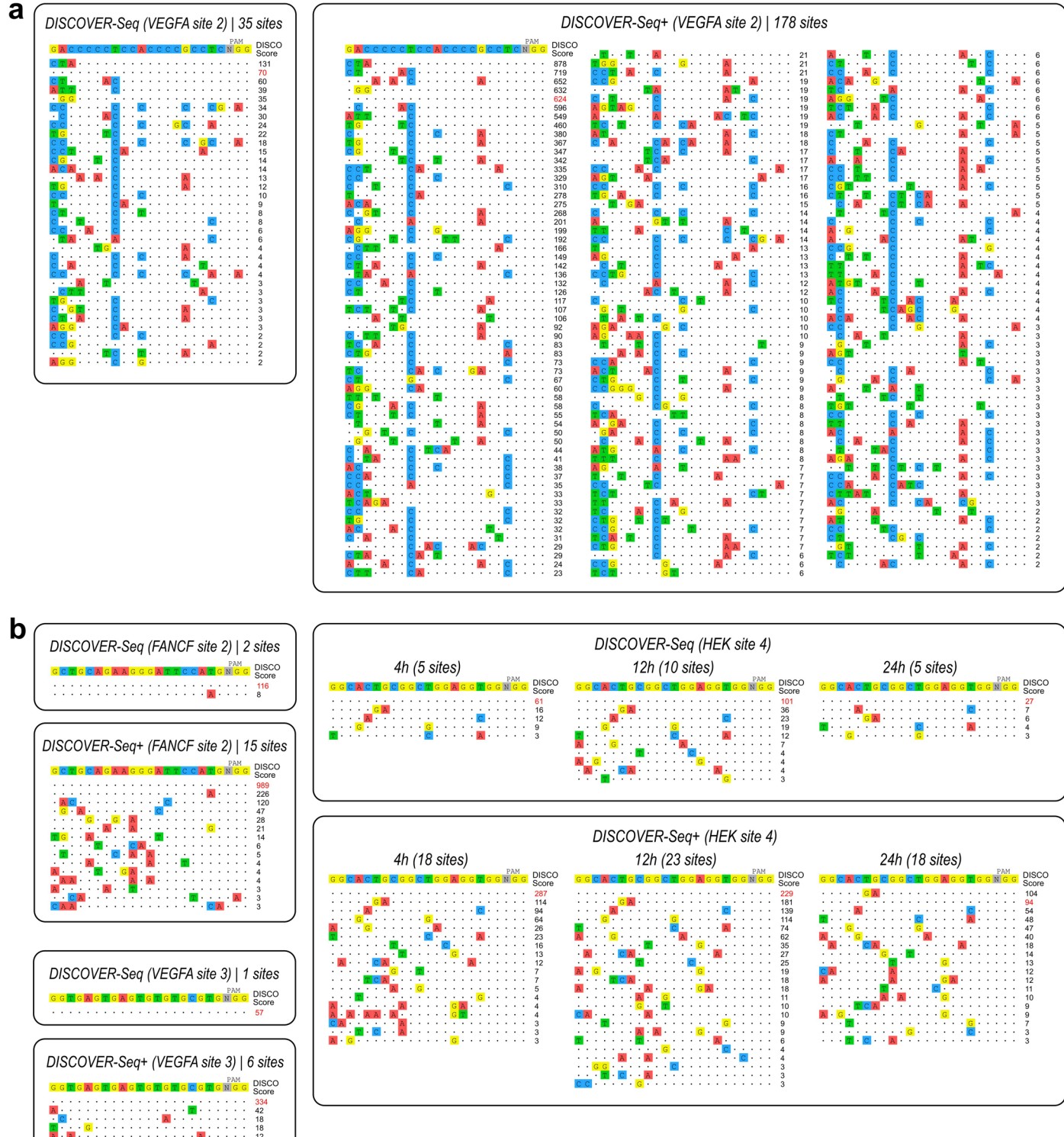

**Extended Data Fig. 2 | Off-target sites discovered by MRE11 ChIP-seq and improved with DNA-PKcs inhibition. a**, *VEGFA site 2* Cas9 target sites detected using DISCOVER-Seq (left) versus DISCOVER-Seq+ (right) in K562 cells. **b**, *FANCF site 2, VEGFA site 3, or HEK site 4* Cas9 target sites detected using DISCOVER-Seq versus DISCOVER-Seq+ in K562 or HEK293T cells, at 4 h, 12 h, or 24 h after Cas9 delivery. Source numerical data are available in source data.

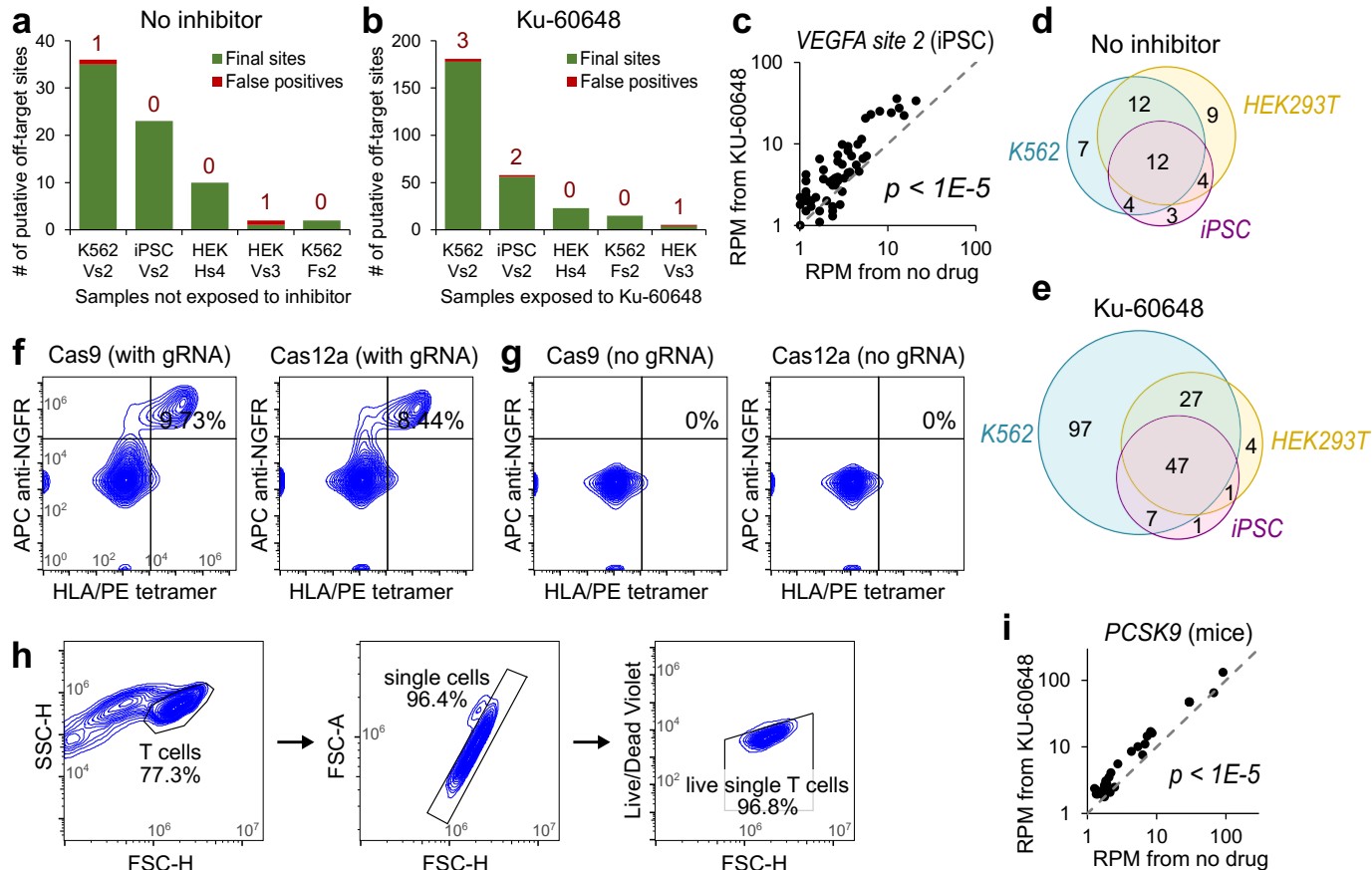

**Extended Data Fig. 3 | Validation of DISCOVER-Seq+ in diverse applications.**
**a-b**, Plot of the total number of initial off-target sites, for samples (a) without
inhibitor and (b) with Ku-60648. Sites labeled as false positive, because they
were also reported in corresponding negative control samples without Cas9, are
colored red with the number of sites indicated above each bar. For x-axis, first
row is cell type, second row is gRNA target (Vs2: VEGFA site 2, Hs4: HEK site 4,
Vs3: VEGFA site 3, Fs2: FANCF site 2). **c**, Plot of MRE11 ChIP-seq enrichment at all
DISCOVER-Seq+ detected target sites within a 1.5 kb window for Cas9 targeting
*VEGFA site 2* in WTC-11 iPSCs. Each point in the plot (55 total) corresponds to a
putative target site. MRE11 enrichment from DISCOVER-Seq+ data (y-axis) versus
DISCOVER-Seq data (x-axis) was significantly different (p = 1.24E-8), determined
using the two-sided Wilcoxon signed-rank test. **d-e**, Venn diagram illustrating
overlap in the *VEGFA site 2* sites identified by (d) DISCOVER-Seq or (e) DISCOVER-
Seq+ in K562 cells (blue), HEK293T cells (yellow), and WTC-11 iPSCs (purple).

**f-g**, (f) Flow cytometry contour plots of T cell populations at day 7 following
CRISPR editing with Cas9 (left) or Cas12a (right). X-axis gates by T cell binding to
a phycoerythrin (PE)-conjugated HLA-A*02:p53 R175H peptide tetramer complex
specific for the tgTCR. Y-axis gates by allophycocyanin (APC)-conjugated
antibody against NGFR (CD271), introduced as part of the CRISPR HDRT as an
editing control. Cells positive for both markers represent successful cancer-
specific tgTCR integration. 9.7% of T cells are positive for both markers using
Cas9 compared to 8.4% with Cas12a. (g) Same, but for negative control samples
without gRNA. **h**, Gating strategy for flow cytometric analysis of live single T
cells, used for panels **f-g. i**, Same as panel **c**, for Cas9 targeting *PCSK9* in the liver
of mice, with 30 total target sites. MRE11 enrichment from DISCOVER-Seq+ was
significantly different (p = 7.90E-6), using the two-sided Wilcoxon signed-rank
test. Source numerical data are available in source data.

# Reporting Summary

Nature Research wishes to improve the reproducibility of the work that we publish. This form provides structure for consistency and transparency in reporting. For further information on Nature Research policies, see our Editorial Policies and the Editorial Policy Checklist.

## Statistics

For all statistical analyses, confirm that the following items are present in the figure legend, table legend, main text, or Methods section.

| n/a | Confirmed | |
|---|---|---|
| ☐ | ☒ | The exact sample size ($n$) for each experimental group/condition, given as a discrete number and unit of measurement |
| ☐ | ☒ | A statement on whether measurements were taken from distinct samples or whether the same sample was measured repeatedly |
| ☐ | ☒ | The statistical test(s) used AND whether they are one- or two-sided<br>*Only common tests should be described solely by name; describe more complex techniques in the Methods section.* |
| ☐ | ☒ | A description of all covariates tested |
| ☐ | ☒ | A description of any assumptions or corrections, such as tests of normality and adjustment for multiple comparisons |
| ☐ | ☒ | A full description of the statistical parameters including central tendency (e.g. means) or other basic estimates (e.g. regression coefficient) AND variation (e.g. standard deviation) or associated estimates of uncertainty (e.g. confidence intervals) |
| ☐ | ☒ | For null hypothesis testing, the test statistic (e.g. $F$, $t$, $r$) with confidence intervals, effect sizes, degrees of freedom and $P$ value noted<br>*Give P values as exact values whenever suitable.* |
| ☒ | ☐ | For Bayesian analysis, information on the choice of priors and Markov chain Monte Carlo settings |
| ☒ | ☐ | For hierarchical and complex designs, identification of the appropriate level for tests and full reporting of outcomes |
| ☐ | ☒ | Estimates of effect sizes (e.g. Cohen's $d$, Pearson's $r$), indicating how they were calculated |

*Our web collection on statistics for biologists contains articles on many of the points above.*

## Software and code

Policy information about availability of computer code

| Data collection | No code was used for collecting the data used in this study. |
|---|---|
| Data analysis | FlowJo v.10, bcl2fastq2, bowtie2, macs2, samtools1.10, scikit-learn1.1, custom code for data analysis (https://github.com/rogerzou/DseqPlus), Blender (version 9cb8a72; Dec 12, 2019) for generating list of off-target sites from sequencing data (https://github.com/staciawyman/blender) |

For manuscripts utilizing custom algorithms or software that are central to the research but not yet described in published literature, software must be made available to editors and reviewers. We strongly encourage code deposition in a community repository (e.g. GitHub). See the Nature Research guidelines for submitting code & software for further information.

## Data

Policy information about availability of data

All manuscripts must include a data availability statement. This statement should provide the following information, where applicable:
- Accession codes, unique identifiers, or web links for publicly available datasets
- A list of figures that have associated raw data
- A description of any restrictions on data availability

Deep-sequencing data that support the findings of this study have been deposited in Sequence Read Archive under BioProject accession PRJNA801688. Sequencing data was analyzed using the hg38 genome assembly (https://www.ncbi.nlm.nih.gov/assembly/GCF_000001405.26). Source data have been provided in Source Data.

# Field-specific reporting

Please select the one below that is the best fit for your research. If you are not sure, read the appropriate sections before making your selection.

☒ Life sciences ☐ Behavioural & social sciences ☐ Ecological, evolutionary & environmental sciences

For a reference copy of the document with all sections, see nature.com/documents/nr-reporting-summary-flat.pdf

# Life sciences study design

All studies must disclose on these points even when the disclosure is negative.

| | |
|---|---|
| Sample size | No statistical method was used to predetermine sample size. |
| Data exclusions | No data was excluded from the analysis. |
| Replication | All attempts at replication were successful (triplicates). |
| Randomization | Allocation was randomly selected. |
| Blinding | Not relevant because no group allocation was performed in this study. |

# Reporting for specific materials, systems and methods

We require information from authors about some types of materials, experimental systems and methods used in many studies. Here, indicate whether each material, system or method listed is relevant to your study. If you are not sure if a list item applies to your research, read the appropriate section before selecting a response.

## Materials & experimental systems

| n/a | Involved in the study |
|---|---|
| ☐ | ☒ Antibodies |
| ☐ | ☒ Eukaryotic cell lines |
| ☒ | ☐ Palaeontology and archaeology |
| ☐ | ☒ Animals and other organisms |
| ☒ | ☐ Human research participants |
| ☒ | ☐ Clinical data |
| ☒ | ☐ Dual use research of concern |

## Methods

| n/a | Involved in the study |
|---|---|
| ☐ | ☒ ChIP-seq |
| ☐ | ☒ Flow cytometry |
| ☒ | ☐ MRI-based neuroimaging |

## Antibodies

| | |
|---|---|
| Antibodies used | rabbit anti-MRE11 (1:750, NB100-142, Novus Biological), mouse anti-BRCA1 (1:500, sc-6954 D9, Santa Cruz Biotechnology), rabbit anti-53BP1 (1:500, ab172580, Abcam), goat anti-Mouse Alexa-594 (1:1000, A-21235, Thermo Fisher), goat anti-Rabbit Atto-647N (1:1000, 40839, Sigma). |
| Validation | - rabbit anti-MRE11 previously validated in HEK293T cells (https://www.science.org/doi/10.1126/science.aav9023).<br>- mouse anti-BRCA1 previously validated in U2OS and HeLa cells (https://datasheets.scbt.com/sc-6954.pdf)<br>- rabbit anti-53BP1 previously validated in 16 studies (https://www.abcam.com/53bp1-antibody-ab172580.html)<br>- goat anti-Mouse Alexa-594 widely validated in over 1200 studies (https://www.thermofisher.com/antibody/product/Goat-anti-Mouse-IgG-H-L-Cross-Adsorbed-Secondary-Antibody-Polyclonal/A-21235)<br>- goat anti-Rabbit Atto-647N validated in over 30 studies (https://www.sigmaaldrich.com/US/en/product/sigma/40839) |

## Eukaryotic cell lines

Policy information about cell lines

| | |
|---|---|
| Cell line source(s) | HEK293T cells (ATCC® CRL-3216™) and K562 cells (ATCC® CCL-243™) are from ATCC. WTC-11 cells are from Coriell Institute (GM25256) |
| Authentication | None of the cell lines were formally authenticated by the authors. The appearance and growth characteristics matched expectations. |
| Mycoplasma contamination | Cell lines tested negative for mycoplasma contamination. |

| Commonly misidentified lines<br>(See ICLAC register) | No commonly misidentified cells were used in this study. |
|---|---|

# Animals and other organisms

Policy information about studies involving animals; ARRIVE guidelines recommended for reporting animal research

| Laboratory animals | 8-10 week old male C57BL/6J mice. |
|---|---|
| Wild animals | No wild animals were used in this study. |
| Field-collected samples | No field-collected samples were used in this study. |
| Ethics oversight | All mouse studies were carried out in accordance with guidelines and approval of the Johns Hopkins University Animal Care and Use Committee (Protocol #MO20M274). |

Note that full information on the approval of the study protocol must also be provided in the manuscript.

# ChIP-seq

## Data deposition

☒ Confirm that both raw and final processed data have been deposited in a public database such as GEO.

☒ Confirm that you have deposited or provided access to graph files (e.g. BED files) for the called peaks.

| Data access links<br>*May remain private before publication.* | Sequencing data: https://www.ncbi.nlm.nih.gov/bioproject/PRJNA801688/.<br>Called peaks from BLENDER (https://github.com/staciawyman/blender) output accessible at: https://github.com/rogerzou/DSeqPlus/tree/main/peaks |
|---|---|
| Files in database submission | List of files displayed on website linked above. |
| Genome browser session<br>(e.g. UCSC) | N/A |

## Methodology

| Replicates | Ranging from 1-5 biological replicates. |
|---|---|
| Sequencing depth | 10 million to 50 million paired-end reads. |
| Antibodies | anti-MRE11 (1:750, NB100-142, Novus Biological) |
| Peak calling parameters | bowtie2 -p 6 -q --local -X 1000 |
| Data quality | Reads were filtered for mapping quality >= 25. Singleton reads, potential PCR duplicates and index reads were removed. |
| Software | bowtie2, macs2, samtools, custom code available at https://github.com/rogerzou/DseqPlus |

# Flow Cytometry

## Plots

Confirm that:

☒ The axis labels state the marker and fluorochrome used (e.g. CD4-FITC).

☒ The axis scales are clearly visible. Include numbers along axes only for bottom left plot of group (a 'group' is an analysis of identical markers).

☒ All plots are contour plots with outliers or pseudocolor plots.

☒ A numerical value for number of cells or percentage (with statistics) is provided.

## Methodology

| Sample preparation | Surface staining for flow cytometry was performed by washing T cells in PBS, pH7.4, followed by staining with  LIVE/DEAD™ Fixable Violet Stain (ThermoFisher) at recommended concentration for 30 minutes on ice in the dark. T cells were then resuspended in Cell Staining Buffer (BioLegend) plus relevant antibodies (APC anti-NGFR [BioLegend]) and PE-conjugated HLA-A*02:p53 R175H tetramer (NIH Tetramer Facility) for 30 minutes at room temperature in the dark. Cells were washed twice in Cell Staining Buffer before resuspension for analysis. |
|---|---|
| Instrument | Flow cytometric analysis was performed on an IntelliCyt iQue Screener PLUS VBR (Sartorius). |

| Software | FlowJo v.10 was used for flow cytometry data analysis. |
|---|---|
| Cell population abundance | A single cell suspension of activated primary human CD3+ T cells was edited by CRISPR/Cas HDR and expanded in the presence or absence of Ku-60648 or vehicle, then subjected to flow cytometric analysis to define the frequency of live single NGFR+/tetramer+ T cells. The abundance of this population was 8.4%-13.3% among CRISPR/Cas-edited and 0% among mock-edited purified primary human CD3+ T cells, respectively. |
| Gating strategy | The gating strategy used for primary human T cells is shown in Extended Data Figure 2. For all experiments, debris was first excluded by a morphology gate based on FSC-H and SSC-H. Non-singlets were excluded from analysis by a single cell gate based on FSC-H and FSC-A. Live cells were selected by gating on cells negative for staining with LIVE/DEAD Fixable Violet dye (405 nm excitation, Invitrogen). Anti-NGFR APC+/tetramer PE+ cells were gated to define successfully edited live, single T cells using appropriate compensation with single-stained controls. |

☒ Tick this box to confirm that a figure exemplifying the gating strategy is provided in the Supplementary Information.

