## [Peer Review File · Nature Methods]

Peer Review Information

Manuscript Title: Improving the sensitivity of in vivo CRISPR off-target detection with DISCOVER-Seq+

Corresponding author name(s): Taekjip Ha

Editorial Notes: n/a

Reviewer Comments & Decisions:

Decision Letter, initial version:

Dear Dr Ha,

Your Brief Communication entitled "Improving the sensitivity of in vivo CRISPR off-target detection with DISCOVER-Seq+" has now been seen by 3 reviewers, whose comments are attached. While they find your work of potential interest, they have raised serious concerns which in our view are sufficiently important that they preclude publication of the work in Nature Methods, at least in its present form.

As you will see, the reviewers raise concerns about the methodology advance DISCOVER-Seq+ represents and its practical utility in detecting in vivo off-target sites.

Should further experimental data allow you to fully address these criticisms we would be willing to look at a revised manuscript (unless, of course, something similar has by then been accepted at Nature Methods or appeared elsewhere). This includes submission or publication of a portion of this work somewhere else. We hope you understand that until we have read the revised paper in its entirety we cannot promise that it will be sent back for peer-review.

If you are interested in revising this manuscript for submission to Nature Methods in the future, please contact me to discuss your appeal before making any revisions. Otherwise, we hope that you find the reviewers' comments helpful when preparing your paper for submission elsewhere.

Best regards,
Lei

Lei Tang, Ph.D.
Senior Editor
Nature Methods

Reviewers' Comments:

Reviewer #1:

Remarks to the Author:

The manuscript by Zou et al. describes the development of the DISCOVER-Seq+ method, an improved technology of previously reported DISCOVER-Seq for detection of Cas9-induced off-targets in vivo. Different from DISCOVER-Seq, one more step was included to treat the cells or animals with chemical inhibitor targeting DNA-PKcs to accumulate MRE11 at CRISPR-targeted sites. This modification increased the detection sensitivity 2- to 5-fold in mouse tissues and immortalized cell lines. As the CRISPR/Cas9 system has been employed for clinical applications, highly sensitive off-target detection methods are very important for the development of CRISPR-based gene therapy.

I agree DISCOVER-Seq+ is a powerful method, especially for in vivo detection of CRISPR-induced off-targets. However, I am not sure whether the improvement over existing DISCOVER-Seq systems with 2-fold increase of sensitivity in vivo is sufficient to publish in Nature Methods.

Major points:

1. Although the authors showed that DNA-PKcs inhibitor Ku-60648 alone did not induce additional DNA damage inside cells through immunofluorescence against 53BP1, more sensitive and quantitative assays should be employed to show whether the inhibitor would induce DNA damage in cell lines as well as in primary cells.
2. In figure 1l, it showed that in cells exposed to Cas9 but without DNA-PKcs inhibition, off-target sites identified by DISCOVER-Seq+ could be detected through targeted deep sequencing. How about the off-target sequencing data in cells treated with DNA-PKcs inhibitor? It would help to demonstrate whether the inhibitor amplified off-targeting effects. Similarly, in figure 1n, the indels were generated with or without DNA-PKcs inhibitor?
3. In cultured cells, DISCOVER-Seq+ is comparable to GUIDE-seq. There is a related question, does the DNA-PKcs inhibitor also increase the sensitivity of GUIDE-seq?
4. The Cas9 system has been used in clinical trials, such as CAR-T and beta-thalassemia. The evaluation of off-target effects, especially the targets used or to be used in clinical trials, in human T cells or hematopoietic stem cells are very critical and important. Since the manuscript showed more off-target sites identified in immobilized cells than iPSCs, is it possible to evaluate 1-2 this kind of sgRNAs in primary human cells?
5. The advantage of DISCOVER-Seq+ is more sensitive than previous method for in vivo study, but DISCOVER-Seq is more practical especially for possible clinical studies. For example, patient biopsies

could be used for DISCOVER-Seq, but it is quite difficult to treat the patients with Ku-60648 before biopsy.

6. It is better to present indel frequencies by amplicon-NGS in Cas9-treated mouse liver samples of both on-target and off-target sites, especially the newly identified sites by DISCOVER-Seq+.

Reviewer #2:

Remarks to the Author:

In this manuscript formatted for Nature Methods Brief Communications, the Authors report an improvement of their previous method DISCOVER-Seq for detecting off-target CRISPR-Cas genome editing activity, which is based on chromatin immunoprecipitation and sequencing (ChIP-seq) using an antibody against the MRE11 repair factor—a key protein involved in the non-homologous recombination pathway that repairs DNA double-strand breaks (DSBs).

The manuscript is well written and the data presented convincingly demonstrate the superiority of DISCOVER-Seq+ over the original method. It remains to be demonstrated whether the rare OTs detected by DISCOVER-Seq+ (as well as by GUIDE-seq) would result in mutagenic events and, therefore, should be regarded as a matter of concern.

Below I list a series of points and suggestions for revision.

MAJOR REMARKS

1) The Authors identify Ku-60648 as the inhibitor yielding the strongest stabilization of MRE11 at DSB sites and therefore the highest increase in the number of OT events detectable by DISCOVER-Seq. The Authors mention that this inhibitor has a good pharmacokinetics, however it is difficult to assess which tissues besides liver (as shown by the Authors) could be effectively targeted with this inhibitor. In general, I suspect that pre-treating cells with the same inhibitor would also enhance OT detection by GUIDE-seq or other cell-based OT-mapping methods such as BLESS/BLISS. Have the Authors tried this approach? The Authors should briefly discuss these points in their manuscript even if it is formatted for Brief Communications.

2) In Fig. 1m, the Authors show that GUIDE-seq detects a sizable number of OTs that are not detected by DISCOVER-Seq+ and vice versa. I remain perplexed about the true nature of these OTs and their potential risk: can it be that these OTs simply represent endogenous DSBs that are erroneously classified as OTs by the OT-detection pipelines used in GUIDE-seq and DISCOVER-Seq(+)? More generally, in which type of chromatin environment do different OTs form? Is there a preference for certain types of

genomic regions? Like in all other papers describing new methods for CRISPR-Cas OT detection, there is no analysis of the chromatin context in which OTs arise, while I think this would be a very valuable information. Can the Authors perform some analyses towards this direction?

3) Many of the OTs identified by DISCOVER-Seq+ (or by GUIDE-Seq) are not associated with indel formation (see, for example, Fig. 1l), raising the possibility that these OT events are not mutagenic and therefore should not be regarded as a major concern. In fact, what really matters is whether some of the recurrent OT events consistently give rise to mutations and/or genomic rearrangements, which can only be ascertained by whole-genome sequencing. The Authors should therefore at least briefly justify the importance of developing yet another OT detection method when it is clear that the mutagenic potential of the identified OTs can only be ruled out using whole genome sequencing. What would be the applicability of DISCOVER-Seq+ in the context of regenerative medicine? I imagine that rather than examining which OTs a sgRNA can generate in vivo it would be more important to perform whole-genome sequencing of DNA from multiple tissues/organs, including those that are not targeted. (For example, in the mouse model presented by the Authors, how can they rule out that mutations/rearrangements would not be induced as a result of spurious Cas9 activity in the liver or other organs?). However, I acknowledge that this is a 'hot' topic of discussion and that multiple complementary approaches are needed to gauge the safety of genome editing techniques.

ADDITIONAL REMARKS

Main text

>>Please provide the name of the test used and whether it is one- or two-tailed, whenever a P value is reported in the manuscript.

>>Page 2, second paragraph from the top (the Authors should have included page numbers to make it easier to point to specific parts): can the Authors add a 1-sentence motivation for why they performed targeted deep sequencing without DNA-PKcs inhibition? (The motivation is clear to this Reviewer but it is left implicit and therefore might not be immediately clear to other Readers).

Figures

>>Fig. 1: it is not clear why the Authors show some results from HEK293T and some from K562: is it because some sgRNAs worked better in one cell line? The Authors should justify in the main text why they show results from one cell line and not from the other or alternatively show results from one cell line in the main figure and from the other cell line in Supplementary Figures. Also, please add the name of the cell line and sgRNA as title in every plot (for instance, in Fig. 1f, g it is not specified which cell line this refers to).

>>Fig. 1c: what does each dot represent? One biological replicate? Is the red line the mean? Please clarify in the legend.

>>Fig. 1e: the legend states n=14 biological replicates, but as I understand this plot pools together results from both HEK293T and K562 cells. Therefore, these are not all biological replicates, but multiple experiments from different cell lines pooled together. The Authors should color differently dots corresponding to HEK293 to distinguish them from dots corresponding to K562 if they want to present the data in the same plot. Also, please clarify what the red lines represent (Mean? Median?).

>>Fig. 1f, g: what does each dot represent? One OT site? Please clarify in the legend. Please also add the number n of how many data points (dots) are shown.

>>Fig. 1h, i: what is MRE11 enrichment? Is it the ratio of the # of reads in the sample vs. CHIP-seq input DNA or control with a secondary Ab only? Please clarify in the legend. Also, why are there two peaks at the on-target site (with the left peak being higher) while this feature is much less pronounced at OT sites? The Authors should comment on this at least in the corresponding figure legend.

>>Fig. 2a, 2j and similar plots in Suppl. Figures: it would be useful to have the number n of OTs displayed on top of each alignment.

>>Fig. 2d: please write the name of the target gene as done in Fig. 1.

>>Fig. 2f, g: are these two examples of OTs? Please clarify. Also, please use a different abbreviation for mismatches as mm will be immediately recognized as the abbreviation for millimeters.

>>Fig. 2h: here the increase in the number of OTs is much less pronounced compared to Fig. 1j: can the Authors provide an explanation for this difference? Is it because the percentage of cells in which Cas9 was successfully delivered is much lower?

Reviewer #3:

Remarks to the Author:

General comments;

The CRISPR-Cas system-derived genome editing tools have revolutionized many research areas including biology and medicine. However, Cas9 nucleases cleave not only on-target site but off-target sites, hindering further therapeutic application. Thus, the profiling of off-target sites in the whole genome is very important especially in the therapeutic application and many methods have been developed up to date via in silico (e.g. Cas-OFFinder, CHOPCHOP, CRISPOR), in vitro (Digenome-seq, SITE-seq, CIRCLE-seq), or in vivo (GUIDE-Seq, BLISS, DISCOVER-Seq) strategies. Following to the GUIDE-Seq, DISCOVER-Seq was developed to determine off-target sites in cells by tracking the precise recruitment of MRE11 to double-strand breaks (DSBs). In this manuscript, the authors demonstrated that inhibition of DNA-PKcs using chemical compounds (Ku-60648 or Nu7026) accumulates MRE11 at CRISPR-targeted sites, resulting in the increase of the DISCOVER-Seq sensitivity, which is named DISCOVER-Seq+. It is of note that the DISCOVER-Seq+ could be applied for in vivo mice simply by the peritoneal injection of Ku-60648, enhancing the value of this method. The novelty of DISCOVER-Seq+ might not be high because it is based on the previous method (DISCOVER-Seq), but this study might be potentially suitable to this

journal, Nat Methods, considering the highly improved sensitivity and unique features for in vivo application.

Specific comments;

1. It seems that the introduction part should be generally improved for broad readers. For example, it will be better to clarify the limitations of the previous methods. Why are other methods limited to restricted cellular systems [6,7], contrast to the present method (DISCOVER-Seq)? That is probably because the GUIDE-Seq or BLISS require the DNA repair process or cell-cycle. Detailed description will be helpful for readers.
2. In the result of Figure 1n, it is acceptable that DISCOVER-Seq+ is better than DISCOVER-Seq in terms of sensitivity because DISCOVER-Seq missed bona-fide off-target sites (OFF1, 2, 4, 5, 7). However, it is confusing to note that DISCOVER-Seq+ is better than GUIDE-Seq because GUIDE-Seq did not miss the bona-fide off-target sites. Rather, the higher sensitivity of DISCOVER-Seq+ than GUIDE-seq sometimes generate higher false-positive data. It should be discussed.
3. It would be very important to confirm that the inhibition of DNA-PKcs does not affect Cas9 activity or editing efficiency. In addition to DNA damage experiments (Fig. S2), it is necessary to compare general editing efficiencies at several target sites with/without Ku-73 60648.
4. It is necessary to provide the editing efficiencies (i.e., indel frequencies) at on-target or off-target sites in Figure 2d-related experiments.
5. It would be better to discuss the limitation of the present method (DISCOVER-Seq+) in discussion part.

Author Rebuttal to Initial comments

Reviewer #1:

Remarks to the Author:

The manuscript by Zou et al. describes the development of the DISCOVER-Seq+ method, an improved technology of previously reported DISCOVER-Seq for detection of Cas9-induced off-targets in vivo. Different from DISCOVER-Seq, one more step was included to treat the cells or animals with chemical inhibitor targeting DNA-PKcs to accumulate MRE11 at CRISPR-targeted sites. This modification increased the detection sensitivity 2- to 5-fold in mouse tissues and immortalized cell lines. As the CRISPR/Cas9 system has been employed for clinical applications, highly sensitive off-target detection methods are very important for the development of CRISPR-based gene therapy.

I agree DISCOVER-Seq+ is a powerful method, especially for in vivo detection of CRISPR-induced off-

targets. However, I am not sure whether the improvement over existing DISCOVER-Seq systems with 2-fold increase of sensitivity in vivo is sufficient to publish in Nature Methods.

We thank the reviewer for their support of our manuscript. Based on the reviewer's helpful feedback, we performed a major revision that includes many new experiments and analyses. First, we demonstrated through complementary assays (including super-resolution STED microscopy) that the likelihood of false positives (such as Ku-60648 alone inducing damage), is low. Second, we performed new control experiments that allowed us to identify and exclude potential false positives sites directly. Third, we evaluated the cellular effects of DNA-PKcs inhibition, finding modulation of repair pathways from NHEJ to MMEJ/HDR. Lastly, we were inspired by the reviewer's suggestion to evaluate our technology in translational contexts such as CAR-T cells, and performed the exact experiment that was suggested – we applied DISCOVER-Seq+ to the knock-in of a therapeutically relevant CAR into primary human T-cells. However, we disagree with the assessment that a 2-fold increase in sensitivity in certain samples is not enough. We often report results as “fold increase in sensitivity” because it is an easily understood relative quantity, but it should not be extrapolated as a measure of absolute success. The reason is that the true number of off-target sites is unknown but finite. As an example, suppose there are in reality 80 total off-target sites and DISCOVER-Seq detects 40 of them while DISCOVER-Seq+ detects 2-fold more, or 80 sites. While the absolute increase in the # of sites is 2-fold, we went from missing half the off-target sites to missing no off-target sites, a vast improvement. In practice, the true number of off-target sites is, of course, unknown, so the bottom line is that our technology achieves greatly superior results compared to previous state-of-the-art with a simple and accessible protocol, which we believe is an accomplishment significant enough for publication in Nature Methods.

Major points:

1. Although the authors showed that DNA-PKcs inhibitor Ku-60648 alone did not induce additional DNA damage inside cells through immunofluorescence against 53BP1, more sensitive and quantitative assays should be employed to show whether the inhibitor would induce DNA damage in cell lines as well as in primary cells.

We thank the reviewer for this important question. In this revision, we now provide multiple lines of evidence to demonstrate that Ku-60648 does not induce additional DNA damage inside cells (i.e., false positives), including in primary cells. Furthermore, we now have an experimental strategy to identify and remove possible false positives; for a given sample, we perform an identical negative control experiment without Cas9. Any site identified in samples exposed to Cas9 that is also found in this negative control without Cas9 is subsequently removed. Together, we accrue numerous pieces of evidence to suggest a low likelihood of Ku-60648 alone inducing DNA damage.

- *No evidence of DNA-damaging properties of Ku-60648 in literature: Ku-60648 is frequently used in prior studies as an inhibitor of DNA-PKcs, either in the experimental setting or as a radiosensitizer for the potential treatment of cancer (<https://www.selleckchem.com/products/ku-0060648.html>). To our knowledge, there has been no prior evidence of any DNA damaging properties of this small molecule drug. The concentration of inhibitor we use is also low at 1 uM in cells.*

- No evidence of DNA damage from Ku-60648 using super-resolution microscopy of DNA repair foci: We agree with the reviewer that epifluorescence confocal microscopy may not be sensitive enough to detect additional DNA damage. To improve detection sensitivity with unbiased quantitative measurement, we used Simulated Emission Depletion (STED) microscopy, a super-resolution microscopy technique. We performed immunofluorescence labeling for 53BP1 and BRCA1, two key signaling molecules involved in major DNA repair pathways (NHEJ vs MMEJ/HDR, respectively), in cells with or without Ku-60648, followed by STED microscopy and automated detection of both 53BP1 and BRCA1 foci. 53BP1 and BRCA1 both form megabase-size foci even in response to a single DNA break, so STED should be more than sufficient to detect any excess DNA damage due to Ku-60648. In both samples with or without Cas9, we do not detect a significant change in the number of DNA repair foci (53BP1 + BRCA1) from exposure to Ku-60648 (**Fig. 1g-i**). Therefore, with remarkably increased sensitivity using super-resolution microscopy and evaluation of another DNA repair foci, we still do not find evidence that Ku-60648 itself induces DNA damage.

Fig. 1g-i

- Only a very limited number of false positive sites are experimentally determined in cells: Whether Ku-60648 itself induces DNA breaks that become detected by DISCOVER-Seq+ is one type of false positive measurement. We performed additional control experiments to evaluate and exclude possible false positive measurements directly. For each experiment with Cas9 (with or without Ku-60648), we performed a corresponding negative control MRE11 ChIP-seq in cells without Cas9. Of the discovered off-target sites in samples with Cas9, we evaluated whether any of those sites are also identified in the corresponding negative control sample. By removing the effect of Cas9, we can evaluate the Ku-60648 effect in isolation. We find that only 1-3 of the discovered off-target sites are also present in the negative control samples, which can therefore be confidently labeled as false positives and excluded (**Fig. S2a-b**). Both DISCOVER-Seq (**Fig. S2a**) and DISCOVER-Seq+ (**Fig. S2b**) led to a small number of these false positive sites. Together, this further suggests that Ku-60648 does not induce DNA damage.

Fig. S2a-b

- For samples with Cas9, genomic regions adjacent to expected Cas9 target sites do not exhibit increased MRE11 enrichment with Ku-60648: At genomic locations 10kb away from expected Cas9 target sites, there is no significant change in MRE11 enrichment (Fig. 2e-f) with use of Ku-60648. This result indicates that MRE11 enrichment is not directly influenced by Ku-60648, providing further evidence that Ku-60648 does not itself lead to DNA damage.

**Fig. 2e-f**

- Samples with Cas9 but without gRNA show no increase in MRE11 enrichment at expected target sites: In primary human T-cells with Cas9 and with gRNA, addition of Ku-60648 led to increased MRE11 enrichment at expected target sites, as anticipated (Fig. 3h). In contrast, in cells with Cas9 but without gRNA, addition of Ku-60648 did not increase MRE11 enrichment at target sites (Fig. 3i). These results further exclude the possibility of Ku-60648 itself inducing DNA damage.

**Fig. 3h-i**

Through this extensive collection of data, we hope it is convincing that Ku-60648 does not itself induce DNA damage that leads to a false positive readout with DISCOVER-Seq+. In the unlikely situation that false positives are present, we also have essential control experiments that can detect and remove such false positive measurements. Our revised manuscript includes all this new data as well as a full paragraph in the Discussion section that extensively discusses all these points.

2. In figure 1l, it showed that in cells exposed to Cas9 but without DNA-PKcs inhibition, off-target sites identified by DISCOVER-Seq+ could be detected through targeted deep sequencing. How about the off-target sequencing data in cells treated with DNA-PKcs inhibitor? It would help to demonstrate whether

the inhibitor amplified off-targeting effects. Similarly, in figure 1n, the indels were generated with or without DNA-PKcs inhibitor?

*We thank the reviewer for raising this question. In the initial figure 1n, the indels were generated without DNA-PKcs inhibitor. We followed the reviewer's fantastic suggestion and proceeded to perform additional experiments to measure indels in samples exposed to DNA-PKcs inhibitor. K562 cells were exposed to Cas9 with FANCF site 2 gRNA, with or without Ku-60648 exposure. 4 days later, we measured indel outcomes using amplicon-NGS at the on-target site and 14 off-target sites found by DISCOVER-Seq+. Exposure to Ku-60648 led to a reduction in +/-1 indels from NHEJ in favor of larger -22 and -24 deletions from MMEJ (**Rebuttal Fig. 1**). We found that exposure to Ku-60648 led to 13 out of 15 target sites having detectable indels with amplicon-NGS (**Rebuttal Fig. 2**), compared to 7 out of 15 target sites with detectable indels without Ku-60648 (**Fig. 2k**). Therefore, the inhibitor amplified off-target effects in targeted sequencing. This may be explained by the fact that MMEJ is a highly error-prone repair pathway [PMID: 28754468, 28101326, 29804829, 33848455]; at off-target sites, especially those experiencing very limited Cas9 activity, having a high indel conversion rate such as with MMEJ is essential to obtain indel efficiencies above the detection limit. In contrast, NHEJ is generally more error-free due to direct ligation of broken DNA – previous studies by our group have estimated the conversion rate of NHEJ to be under 15% at a target site [PMID: 32527834].*

*We decided to not feature Rebuttal Fig. 1 or 2 in the manuscript for the following reason: amplicon sequencing was mainly employed in our study to determine if some off-target sites newly discovered with DISCOVER-Seq+ have evidence of mutations with standard CRISPR genome editing. The answer is yes, which is the focus of a paragraph in the Results section. We already know that DNA-PKcs inhibition changes repair outcomes, so also including amplicon sequencing data in cells with DNA-PKcs inhibition is not relevant to the main message and would only detract from that section. Furthermore, we already included a new paragraph early in the Results section that experimentally explores the effect of DNA-PKcs inhibition on DNA repair pathways, clearly showing modulation from NHEJ to MMEJ/HDR (**Fig. 1g-j**).*

Rebuttal Fig. 1

Rebuttal Fig. 2

Fig. 2k

3. In cultured cells, DISCOVER-Seq+ is comparable to GUIDE-seq. There is a related question, does the DNA-PKcs inhibitor also increase the sensitivity of GUIDE-seq?

DNA-PKcs inhibitors such as Ku-60648 block NHEJ, and GUIDE-seq relies on NHEJ to incorporate a small sequence into the genome for amplification [PMID: 25513782]. Therefore, it would be reasonable to hypothesize that DNA-PKcs inhibition would reduce the performance of GUIDE-seq. This great question by the reviewer does raise the important consideration of whether other approaches to inhibit DNA repair have the potential to improve off-target detection. We now discuss this point in the Discussion section and emphasize how that is an important area of future study.

4. The Cas9 system has been used in clinical trials, such as CAR-T and beta-thalassemia. The evaluation of off-target effects, especially the targets used or to be used in clinical trials, in human T cells or hematopoietic stem cells are very critical and important. Since the manuscript showed more off-target sites identified in immobilized cells than iPSCs, is it possible to evaluate 1-2 this kind of sgRNAs in primary human cells?

We thank the reviewer for this important suggestion. In this resubmission, we perform the exact experiment suggested by the reviewer. We applied DISCOVER-Seq+ ex vivo to knock-in of a chimeric antigen receptor (CAR) construct into primary human T-cells [PMID: 29567707]. We electroporated Cas9 targeting TRA (T Cell Receptor Alpha Locus) along with a 4699 bp homology-directed repair template (HDRT) encoding a CAR specific for the HLA receptor loaded with an R175H mutated p53 peptide, then performed DISCOVER-Seq+ 12 hours later (Fig. 3d). The specific R175H mutation that is targeted by the CAR is the most prevalent p53 gain-of-function mutation in human cancers [PMID: 34439241]. DISCOVER-Seq+ identified 20 off-target sites genome-wide compared to 4 with DISCOVER-Seq (Fig. 3e), and led to significantly greater MRE11 enrichment at all discovered sites (Fig. 3f-h). In contrast, samples without Cas9 exhibited no change in enrichment with Ku-60648, further confirming that the inhibitor alone does not induce damage (Fig. 3i). DISCOVER-Seq+ also has potential to compare off-target profiles between different types of CRISPR nucleases. As a proof of concept, we compared the performance of Cas9 to Cas12a (Cpf1), targeting the same position in TRA. DISCOVER-Seq+ at 12 hours after Cas12a editing only identified the on-target site and no off-target sites (Fig. 3j), consistent with the improved specificity of Cas12a. Flow cytometry for CAR expression after 7 days in T-cells without Ku-60648 exposure showed CAR integration efficiencies of 8.44% for Cas12a and 9.73% for Cas9 (Fig. S2d-e). Together, our preliminary

analysis using DISCOVER-Seq+ revealed that use of Cas12a maintained adequate CAR integration while greatly reducing undesirable off-target damage. Furthermore, these experiments demonstrated that DISCOVER-Seq+ is directly compatible with CRISPR knock-in using a homology template in primary human T-cells. Most importantly, these experiments demonstrate the utility of DISCOVER-Seq+ in identifying potentially harmful off-target sites that would otherwise go undetected in emerging human therapies.

Fig. 4d-j

Fig. 52d-e

5. The advantage of DISCOVER-Seq+ is more sensitive than previous methods for in vivo study, but DISCOVER-Seq is more practical especially for possible clinical studies. For example, patient biopsies could be used for DISCOVER-Seq, but it is quite difficult to treat the patients with Ku-60648 before biopsy.

We believe there are very limited scenarios where off-target detection would require systemic Ku-60648 delivery into patients. Once CRISPR is delivered to patients, it is too late to undo. Rather, the role of methods like DISCOVER-Seq+ is to validate gRNAs by detecting off-target editing with high sensitivity before genome editing is performed on human subjects. For example, our technology can be applied to translational model systems such as mice and non-human primates to evaluate the safety of potential CRISPR therapeutics prior to their clinical implementation.

In addition, many clinical applications of CRISPR utilize ex vivo systems such as CAR-T and hematopoietic stem cells, which the reviewer has mentioned. DISCOVER-Seq+ is directly compatible with these applications, where a subset of cells can be exposed to Ku-60648 for off-target detection. If the off-target profile looks good, then the remainder of cells (without Ku-60648 exposure) can be infused into the patient

as a therapeutic. In the Discussion section, we now include a paragraph that describes possible clinical uses cases of DISCOVER-Seq+.

6. It is better to present indel frequencies by amplicon-NGS in Cas9-treated mouse liver samples of both on-target and off-target sites, especially the newly identified sites by DISCOVER-Seq+.

*We thank the reviewer for raising this important topic. While measuring indel frequencies by amplicon-NGS in Cas9-treated mouse liver samples is feasible, we believe that it is relatively redundant in light of our new results in cell lines, and would not add significantly more to this manuscript. This is because as we showed in our revised manuscript, amplicon-NGS is a sub-optimal strategy for off-target detection. Because not all double strand cut events result in mutagenesis in a given experiment, it is more important to determine the totality of double strand cuts (the strength of DISCOVER-Seq+) rather than only those that produce mutagenesis. In agreement with this, in **Fig. 2k-l**, we showed there is a subset of off-target sites found by DISCOVER-Seq+ that did not have evidence of indels via amplicon-NGS. Additionally, we have new, strong evidence that confirms that these sites without indels are not false positive sites (**Fig. S2a-b**). Together, these results lead to the conclusion that all reported DISCOVER-Seq+ off-target sites are true positives, thereby confirming that amplicon-NGS is less sensitive since it fails to detect evidence of mutations at many off-target sites. Additional lines of evidence support this conclusion:*

- (1) A target site can experience DSBs but not lead to mutagenesis. Indeed, the rate of indel conversion after a DSB may not be high, estimated to be under 15% at a single site as shown in our recent study [PMID: 32527834].*
- (2) While it is important to identify mutagenesis, DSBs have deleterious effect even when adequately repaired and must therefore be identified. Indeed, recent evidence by our group showed that even a single DSB can inhibit cell division and induce local epigenetic changes such as increased accessibility [PMID: 36064968].*
- (3) Amplicon sequencing is critically underequipped to detect rare events. Amplicon sequencing does not enrich for Cas9-manipulated sites – assuming no amplification bias, amplicon-NGS reports on the ‘true’ mutation percentage at a given off-target site within its limit of detection, which can be approximated to be ~0.01% [PMID: 25513782]. By rough estimate, this means that in a mouse liver estimated to have approximately 300 million cells, more than 30,000 cells need to have a mutation at both alleles at a specific site to be detected by amplicon-NGS ($0.0001 * 3E8$), assuming adequate sequencing depth. In contrast, dedicated off-target detection methods such as DISCOVER-Seq+ specifically enrich for DNA fragments that are manipulated by CRISPR nucleases. For DISCOVER-Seq+, this is achieved by selectively amplifying MRE11-bound fragments; for GUIDE-seq, by selectively amplifying positions with double-strand oligodeoxynucleotide (dsODN) incorporated into genomic DNA; for BLISS/BLESS, by selectively amplifying positions adjacent to DSBs. These strategies specifically target CRISPR-manipulated sites, thereby increasing sensitivity while minimizing sequencing depth.*

*Therefore, there can be a very low level of CRISPR activity at an off-target site that does not get detected by amplicon-NGS but is detected by DISCOVER-Seq+, suggesting that amplicon-NGS is not great at detecting CRISPR off-target activity. Given that we have performed the complete set of amplicon-NGS for K562 FANCF site 2 and obtained interesting results presented in **Fig. 2k-l** and in response to a previous question by this reviewer, we believe that performing the same experiment on Cas9-treated mouse liver*

samples, while possible, is relatively redundant and would not add significantly more to this manuscript. We hope that the reviewer appreciates the new experiments and insights on these topics in our greatly expanded manuscript.

Reviewer #2:

Remarks to the Author:

In this manuscript formatted for Nature Methods Brief Communications, the Authors report an improvement of their previous method DISCOVER-Seq for detecting off-target CRISPR-Cas genome editing activity, which is based on chromatin immunoprecipitation and sequencing (ChIP-seq) using an antibody against the MRE11 repair factor—a key protein involved in the non-homologous recombination pathway that repairs DNA double-strand breaks (DSBs).

The manuscript is well written and the data presented convincingly demonstrate the superiority of DISCOVER-Seq+ over the original method. It remains to be demonstrated whether the rare OTs detected by DISCOVER-Seq+ (as well as by GUIDE-seq) would result in mutagenic events and, therefore, should be regarded as a matter of concern.

We thank this reviewer for their very helpful feedback. We performed major revisions of our work based on this feedback, including many new experiments and analyses. First, we demonstrated through complementary assays (including super-resolution STED microscopy) that the likelihood of false positives is low. Second, we performed new control experiments that allowed us to directly identify and exclude potential false positives. Third, we evaluated the cellular effects of DNA-PKcs inhibition, finding modulation of repair pathways from NHEJ to MMEJ/HDR. Lastly, we applied our technology to the knock-in of a therapeutically relevant chimeric antigen receptor (CAR) into primary human T-cells.

We would also like to clarify that the goal of our study is to detect off-target CRISPR activity, regardless of whether they result in mutagenic events. DNA damage, especially double-stranded breaks, induces local epigenetic changes and can perturb cell division and transcription regardless of adequate repair to the wild-type sequence. Furthermore, when a mutagenic event does not occur and the DNA is repaired faithfully, that results in further rounds of DNA damage and repair. Therefore, DNA damage itself is already a matter of concern, and a major advantage of our method is that it determines positions where CRISPR induces unwanted DNA damage inside a cell, not only the subset with detectable mutagenesis.

Below I list a series of points and suggestions for revision.

MAJOR REMARKS

1) The Authors identify Ku-60648 as the inhibitor yielding the strongest stabilization of MRE11 at DSB sites and therefore the highest increase in the number of OT events detectable by DISCOVER-Seq. The Authors mention that this inhibitor has a good pharmacokinetics, however it is difficult to assess which tissues besides liver (as shown by the Authors) could be effectively targeted with this inhibitor.

We thank the reviewer for this important comment. As a small molecule compound with a molecular weight of 582.71 g/mol, Ku-60648 is expected to be well-distributed throughout the body. Indeed, the pharmacokinetics of Ku-60648 has been previously characterized in mice with tumor xenografts. Both p.o. and i.p. routes result in bioavailability > 75%, and a half-life of around 2 hours in serum. Furthermore, the

drug was found to effectively penetrate subcutaneous tumor xenografts in their model system [PMID: 22576130].

We would be eager to experimentally confirm the efficacy of this inhibitor in other tissues. However, the biggest barrier to such an experiment is a convenient strategy for delivery of Cas9 *in vivo* to these tissues. The majority of known delivery methods (including LNPs and adenovirus), most reliably deliver Cas9 to the liver. However, this is an active area of research and we anticipate that our method would become applicable as other organs become accessible for Cas9 delivery as recent studies indicate [PMID: 32051598]. We agree that delivery, whether of the small-molecule inhibitor or the CRISPR system itself, is a very important point, and include a discussion about this important topic in the Discussion section.

In general, I suspect that pre-treating cells with the same inhibitor would also enhance OT detection by GUIDE-seq or other cell-based OT-mapping methods such as BLESS/BLISS. Have the Authors tried this approach? The Authors should briefly discuss these points in their manuscript even if it is formatted for Brief Communications.

We thank the reviewer for this interesting comment. We verify through new experiments in this revision – through (1) super-resolution imaging of DNA repair foci associated with specific repair pathways and (Fig. 1g-i) (2) sequencing of repair outcomes (Fig. 1j) – of samples exposed to Cas9 with or without DNA-PKcs inhibition, that Ku-60648 inhibits repair through the NHEJ pathway. Because GUIDE-seq relies on NHEJ to incorporate a double-strand oligodeoxynucleotide (dsODN) sequence [PMID: 25513782], we believe that off-target detection would actually be reduced with the DNA-PKcs inhibitor. In contrast, methods like BLESS/BLISS [PMID: 28497783] may exhibit improved off-target detection because the fast NHEJ repair pathway is inhibited in favor of slower MMEJ/HDR repair, so the quantity of DSBs at a given moment is likely increased. We agree that this is an important point and include a discussion of this topic in the Discussion section, including whether other approaches to inhibit DNA repair have the potential to improve off-target detection.

Fig. 1g-i

Fig. 1j

2) In Fig. 1m, the Authors show that GUIDE-seq detects a sizable number of OTs that are not detected by DISCOVER-Seq+ and vice versa. I remain perplexed about the true nature of these OTs and their potential risk: can it be that these OTs simply represent endogenous DSBs that are erroneously classified as OTs by the OT-detection pipelines used in GUIDE-seq and DISCOVER-Seq(+)? More generally, in which type of chromatin environment do different OTs form? Is there a preference for certain types of genomic regions? Like in all other papers describing new methods for CRISPR-Cas OT detection, there is no analysis of the chromatin context in which OTs arise, while I think this would be a very valuable information. Can the Authors perform some analyses towards this direction?

We thank the reviewer for this very helpful comment. We agree that our original submission failed to exclude the possibility that some extra off-targets may simply represent endogenous DSBs (i.e., false positives). Inspired by this important feedback, in this revision, we performed additional control experiments to evaluate and exclude possible false positive measurements directly. For each experiment with Cas9 (with or without Ku-60648), we performed a corresponding negative control MRE11 ChIP-seq in cells without Cas9. Of the discovered off-target sites in samples with Cas9, we evaluated whether any of those sites are also identified in the corresponding negative control sample. We find that only 1-3 of the discovered off-target sites are also present in the negative control samples, which are confidently labeled as false positives and can be excluded (Fig. S2a-b). The very few false positives are found with both DISCOVER-Seq (Fig. S2a) and DISCOVER-Seq+ (Fig. S2b). The revision now also includes an extensive paragraph in the Discussion section that reiterates these findings, in addition to summarizing other evidence in this resubmission that help exclude further possibility of false positives.

**Fig. S2a-b**

We thank the reviewer for their fantastic suggestion to study the genomic regions of identified off-target sites. There have been previous studies of off-target detection methods that investigate this question. Notably, the original GUIDE-seq paper (PMID: 25513782) tabulated the target locations of promiscuous gRNAs (that we also use in this study), finding sites in intergenic, exons, and intronic regions.

Target	Intergenic	Exons	Introns
VEGFA site 1	8	1	12
VEGFA site 2	39	24	88
VEGFA site 3	19	6	34
EMX1	9	1	5
FANCF	3	3	2
RNF2	0	0	0
HEK 293 site 1	6	1	2
HEK 293 site 2	1	0	1
HEK 293 site 3	2	1	2
HEK 293 site 4	39	20	74

A later paper (PMID: 32541958) directly compared the results of their *in vitro* method (CHANGE-seq) to GUIDE-seq, which is performed in cells; this allowed the authors to determine off-target bias when CRISPR editing is performed in cells, which can most likely be attributed to chromatin context. Indeed, they found that off-target sites located in a region with high gene expression were more frequently found in the cellular method (GUIDE-seq) compared to the *in vitro* method (CHANGE-seq), suggesting that Cas9 off-target activity is enriched in active chromatin states.

Fig. 5a-b of CHANGE-seq manuscript, showing that Cas9 off-target activity is enriched in active chromatin states.

For our submission, because we use many of the same gRNAs as previous studies, we believe that re-performing the analysis of chromatin context of off-target sites that have already been evaluated in prior works will likely not contribute further insights.

Finally, we have experimentally determined that between different cell types but with the same Cas9 gRNA, there are differences between identified off-target sites. For example, for the VEGFA site 2 gRNA, the off-target sites discovered between different cell types (K562 vs HEK293T vs iPSC) have strong overlap, but also show notable differences (Fig. S2g-h). We hypothesize that these cell-type differences can be attributed to epigenetic differences between different cell types. These results further emphasize the versatility featured by DISCOVER-Seq+, which is compatible with and can evaluate off-target editing directly in the model system of interest. We now include an extended discussion of these points in the Discussion section of the revised manuscript.

Fig. S2g-h

3) Many of the OTs identified by DISCOVER-Seq+ (or by GUIDE-Seq) are not associated with indel formation (see, for example, Fig. 1l), raising the possibility that these OT events are not mutagenic and therefore should not be regarded as a major concern. In fact, what really matters is whether some of the recurrent OT events consistently give rise to mutations and/or genomic rearrangements, which can only be ascertained by whole-genome sequencing. The Authors should therefore at least briefly justify the importance of developing yet another OT detection method when it is clear that the mutagenic potential of the identified OTs can only be ruled out using whole genome sequencing. What would be the applicability of DISCOVER-Seq+ in the context of regenerative medicine? I imagine that rather than examining which OTs a sgRNA can generate in vivo it would be more important to perform whole-genome sequencing of DNA from multiple tissues/organs, including those that are not targeted. (For example, in the mouse model presented by the Authors, how can they rule out that mutations/rearrangements would not be induced as a result of spurious Cas9 activity in the liver or other organs?). However, I acknowledge that this is a ‘hot’ topic of discussion and that multiple complementary approaches are needed to gauge the safety of genome editing techniques.

We thank the reviewer for raising these important points.

The reason why whole genome sequencing (WGS) cannot reliably detect off-target sites is a matter of genomic coverage. Due to the size of the human genome, 100x coverage of the genome would be considered a very successful WGS experiment. By rough approximation, that means from a pool of 1 million cells, WGS has the sensitivity to only detect a specific indel if it is found in approximately 10,000 cells (1%). However, off-target sites are generally found at much lower frequencies. These rare events are crucial to identify since a small number of cells can result in tumorigenesis, for example. This limitation of WGS is why there is great interest in designing off-target detection methods that include an enrichment step to improve the chances of finding off-target sites over direct sequencing. For example, the enrichment step of GUIDE-seq is to selectively amplify genomic DNA that incorporates a dsODN sequence as a result of CRISPR activity. The enrichment step of BLESS/BLISS is to selectively amplify only DNA around DSBs. For DISCOVER-Seq and DISCOVER-Seq+, our enrichment step is to selectively amplify only DNA that is bound to MRE11, given that MRE11 localizes to DNA damaged by CRISPR.

Even non-mutagenic OT events are a major concern. We disagree with the assessment that mutations are needed for a DSB event to be a major concern. Recent evidence by our group showed that even a single DSB can inhibit cell division and induce local epigenetic changes such as increased accessibility [PMID: 36064968]. Also, failure to detect mutagenesis events in a given experiment does not exclude the possibility of mutagenesis in the whole organism – for example, there are estimated to be 300 million cells

in the mouse liver alone; if one cell exhibits a deleterious mutation, it can only be detected if that specific cell's DNA gets amplified and sequenced with extremely high depth. However, that site could still be detected by DISCOVER-Seq+ since it was likely damaged in more cells than the one that failed to repair it adequately. In summary, absence of detectable indels by sequencing does not mean mutagenesis does not exist; alternative assays such as DISCOVER-Seq+ or GUIDE-seq may be better at finding evidence of their existence. Further, even if hypothetically mutagenesis does not occur, the existence of off-target DNA damage alone is worth identifying and minimizing. In the Discussion section, we now include a detailed paragraph that explains why indel measurements are sub-optimal to evaluate for CRISPR off-target sites, and why evaluation of DNA damage, rather than DNA mutagenesis, is preferred for CRISPR off-target detection.

Together, we believe this discussion supports the development and use of DISCOVER-Seq+. In our revised manuscript, we included many new experiments, analyses, and discussions to further support this claim.

ADDITIONAL REMARKS

We greatly appreciate the reviewer's detailed examination of our manuscript for small errors that have gone unnoticed. Fixing these errors have greatly improved our manuscript.

Main text

>>Please provide the name of the test used and whether it is one- or two-tailed, whenever a P value is reported in the manuscript.

The name and number of tails/sides of all statistical testing is now reported in the main text as well as figure legends.

>>Page 2, second paragraph from the top (the Authors should have included page numbers to make it easier to point to specific parts): can the Authors add a 1-sentence motivation for why they performed targeted deep sequencing without DNA-PKcs inhibition? (The motivation is clear to this Reviewer but it is left implicit and therefore might not be immediately clear to other Readers).

We thank the reviewer for this insightful comment. The manuscript is now expanded so that motivations for various experiments are better explained, including this one.

Figures

>>Fig. 1: it is not clear why the Authors show some results from HEK293T and some from K562: is it because some sgRNAs worked better in one cell line? The Authors should justify in the main text why they show results from one cell line and not from the other or alternatively show results from one cell line in the main figure and from the other cell line in Supplementary Figures. Also, please add the name of the cell line and sgRNA as title in every plot (for instance, in Fig. 1f, g it is not specified which cell line this refers to).

We paired specific gRNA with specific cell types arbitrarily, because evaluating all combinations of cell lines and gRNAs would be beyond the scope of this manuscript. We chose to include all our findings in the main

figures (rather than supplemental figures) because we know editors prefer them as such. In our revised manuscript, we add more detail about the specific gRNA and cell type used for each experiment. The name of each cell line and sgRNA are now added to every relevant plot.

>>Fig. 1c: what does each dot represent? One biological replicate? Is the red line the mean? Please clarify in the legend.

We now clarify in the legend that: "Each point corresponds to a different biological replicate of a sample exposed to the DNA repair inhibitor listed in the x-axis. Red line is mean of two biological replicates."

>>Fig. 1e: the legend states n=14 biological replicates, but as I understand this plot pools together results from both HEK293T and K562 cells. Therefore, these are not all biological replicates, but multiple experiments from different cell lines pooled together. The Authors should color differently dots corresponding to HEK293T to distinguish them from dots corresponding to K562 if they want to present the data in the same plot. Also, please clarify what the red lines represent (Mean? Median?).

We thank the reviewer for this great suggestion. Each dot is now color labeled with a figure legend so that the proper experiment is referenced. The legend is now modified, removing "biological replicates" and replacing with "n = 14 total samples pooled from panel d". The red line represents mean, and is now clearly indicated in the figure legend.

Fig. 1e

>>Fig. 1f, g: what does each dot represent? One OT site? Please clarify in the legend. Please also add the number n of how many data points (dots) are shown.

We thank the reviewer for this detailed comment. The legend for each of these plots now includes what the dots represent (a different target site), as well as the number of data points.

>>Fig. 1h, i: what is MRE11 enrichment? Is it the ratio of the # of reads in the sample vs. ChIP-seq input DNA or control with a secondary Ab only? Please clarify in the legend. Also, why are there two peaks at the on-target site (with the left peak being higher) while this feature is much less pronounced at OT sites? The Authors should comment on this at least in the corresponding figure legend.

We thank the reviewer for raising the possibility of readers being unfamiliar with how MRE11 enrichment is defined, and the shape of such enrichment. The definition of MRE11 enrichment is now explicitly stated in the main text and the figure legend. Furthermore, we now explain the shape of MRE11 enrichment in the main manuscript, with an illustration included in **Fig. S1a** (copied below). As now explained in the manuscript, the shape of the peaks is well-defined, first introduced in prior studies, and reflects the region of the enriched fragments flanking the cut site that were sequenced with paired-end Illumina sequencing.

Fig. S1a

>>Fig. 2a, 2j and similar plots in Suppl. Figures: it would be useful to have the number n of OTs displayed on top of each alignment.

All plots of this form now have the number of target sites displayed in the plot header.

>>Fig. 2d: please write the name of the target gene as done in Fig. 1.

All plots now have the target gene written.

>>Fig. 2f, g: are these two examples of OTs? Please clarify. Also, please use a different abbreviation for mismatches as mm will be immediately recognized as the abbreviation for millimeters.

Yes, they are examples of OTs, we apologize for the ambiguity. All abbreviations of 'mm' have now been removed and replaced with 'mismatch'.

>>Fig. 2h: here the increase in the number of OTs is much less pronounced compared to Fig. 1j: can the Authors provide an explanation for this difference? Is it because the percentage of cells in which Cas9 was successfully delivered is much lower?

We thank the reviewer for this detailed observation. We included a statement in the Discussion that addresses this point, in addition to other limitations. We hypothesize it may be due to the reduced effective concentration of inhibitor or Cas9 in the liver. Further optimization of drug dosing and Cas9 delivery may improve outcomes.

Reviewer #3:

Remarks to the Author:

General comments;

The CRISPR-Cas system-derived genome editing tools have revolutionized many research areas including biology and medicine. However, Cas9 nucleases cleave not only on-target site but off-target sites, hindering further therapeutic application. Thus, the profiling of off-target sites in the whole genome is very important especially in the therapeutic application and many methods have been developed up to date via in silico (e.g. Cas-OFFinder, CHOPCHOP, CRISPR), in vitro (Digenome-seq, SITE-seq, CIRCLE-seq), or in vivo (GUIDE-Seq, BLISS, DISCOVER-Seq) strategies. Following to the GUIDE-Seq, DISCOVER-Seq was developed to determine off-target sites in cells by tracking the precise recruitment of MRE11 to double-strand breaks (DSBs). In this manuscript, the authors demonstrated that inhibition of DNA-PKcs using chemical compounds (Ku-60648 or Nu7026) accumulates MRE11 at CRISPR-targeted sites, resulting in the increase of the DISCOVER-Seq sensitivity, which is named DISCOVER-Seq+. It is of note that the DISCOVER-Seq+ could be applied for in vivo mice simply by the peritoneal injection of Ku-60648, enhancing the value of this method. The novelty of DISCOVER-Seq+ might not be high because it is based on the previous method (DISCOVER-Seq), but this study might be potentially suitable to this journal, Nat Methods, considering the highly improved sensitivity and unique features for in vivo application.

We thank the reviewer for their support of our manuscript. Novelty is of course subjective, but as the reviewer states, the superior results speak for themselves. While this method is derived from a previous method, we intentionally note similarities to emphasize that part of the novelty is in the simplicity itself - through one intervention guided by a detailed understanding of Cas9-mediated DNA damage and repair, we are able to greatly boost the sensitivity of off-target detection, even in vivo. We believe that use of a DNA-PKcs inhibitor is creative and non-obvious as we had to screen through several DNA repair inhibitors to identify it, and the fact that the method is simple is a great virtue for it will be much easier to reproduce and adopt across the field. It would be reasonable to argue that our method is superior to a hypothetical method that yields the identical outcome but is more difficult to execute. Therefore, we believe this vision for our method strongly aligns with the scope of Nature Methods.

Our revision includes many new experiments and analyses that address all reviewer comments. Notably, we extensively characterized the possibility of false positives, including a new approach to directly identify potential false positive sites. In addition, we evaluated the cellular effects of DNA-PKcs inhibition, finding modulation of repair pathways from NHEJ to MMEJ/HDR. Finally, we applied our technology to a new translatable application – knock-in of a chimeric antigen receptor (CAR) into primary human T-cells. This application highlights the pressing need for sensitive evaluation of off-target sites since each patient receives a bespoke T-cell line derived from their own cells; thus a precision-medicine approach will be required to ensure each patient's off-target profile is acceptable before therapy. We also expanded the main text, including the introduction and discussion sections, to ensure more complete explanations.

Specific comments;

1. It seems that the introduction part should be generally improved for broad readers. For example, it will be better to clarify the limitations of the previous methods. Why are other methods limited to restricted cellular systems [6,7], contrast to the present method (DISCOVER-Seq)? That is probably because the GUIDE-Seq or BLISS require the DNA repair process or cell-cycle. Detailed description will be helpful for readers.

We thank the reviewer for this great suggestion. We added more detail in the introduction and discussion sections to address comparisons to previous methods and why previous methods are lacking in certain performance characteristics.

2. In the result of Figure 1n, it is acceptable that DISCOVER-Seq+ is better than DISCOVER-Seq in terms of sensitivity because DISCOVER-Seq missed bona-fide off-target sites (OFF1, 2, 4, 5, 7). However, it is confusing to note that DISCOVER-Seq+ is better than GUIDE-Seq because GUIDE-Seq did not miss the bona-fide off-target sites. Rather, the higher sensitivity of DISCOVER-Seq+ than GUIDE-seq sometimes generate higher false-positive data. It should be discussed.

We thank the reviewer for these important comments. In terms of sensitivity in immortalized cells, we believe our results suggest DISCOVER-Seq+ and GUIDE-seq are comparable (Fig. 2I). However, we argue in the main text that DISCOVER-Seq+ is better than GUIDE-Seq in terms of applicability: DISCOVER-Seq+ is directly applicable to in vivo systems (Fig. 4) and in primary cells such as knock-in of a CAR construct into primary human T-cells (Fig. 3), an additional application we included in this revision. Notably, GUIDE-seq is not compatible with either of these clinically translatable applications [PMID: 25513782].

Regarding false positives, we agree that our original study failed to exclude the possibility that some off-target sites may be false positives. In this revision, we devised additional control experiments that can evaluate and exclude possible false positive sites directly. For each experiment with Cas9 (with or without Ku-60648), we performed a corresponding negative control MRE11 ChIP-seq in cells without Cas9. Of the discovered off-target sites in samples with Cas9, we evaluated whether any of those sites are also identified in the corresponding negative control sample. We find that only 1-3 of the discovered off-target sites are also present in the negative control samples, which are confidently labeled as false positives and can be excluded (Fig. S2a-b). The very few false positives are found with both DISCOVER-Seq (Fig. S2a) and DISCOVER-Seq+ (Fig. S2b). The revision now also includes an extensive paragraph in the Discussion section that reiterates these findings, in addition to summarizing other evidence in this resubmission that help exclude further possibility of false positives.

Fig. S2a-b

3. It would be very important to confirm that the inhibition of DNA-PKcs does not affect Cas9 activity or editing efficiency. In addition to DNA damage experiments (Fig. S2), it is necessary to compare general editing efficiencies at several target sites with/without Ku-60648.

We thank the reviewer for bringing up these important points of discussion. We performed new experiments to directly evaluate the effect of DNA-PKcs on DNA repair after CRISPR damage, as well as determined editing efficiencies and indel outcomes at target-sites with vs. without Ku-60648.

It is clear that inhibition of DNA-PKcs affects cellular DNA repair, and therefore editing outcomes. We first performed immunofluorescence labeling for 53BP1 and BRCA1, two key signaling molecules involved in major DNA repair pathways (NHEJ vs MMEJ/HDR, respectively), in U2OS cells exposed to Cas9 with a gRNA targeting over 100 sites in the genome. For the readout, we used Simulated Emission Depletion (STED) microscopy, a super-resolution microscopy technique, to detect both 53BP1 and BRCA1 foci. 53BP1 and BRCA1 both form megabase-size foci even in response to a single DNA break, which should be more than sufficient to detect using STED. After Cas9 damage, we found that DNA-PKcs inhibition using Ku-60648 led to a significant reduction in 53BP1 foci relative to BRCA1, consistent with suppression of NHEJ in favor of HDR/MMEJ ($p < 1E-4$; two-sided Wilcoxon rank sum test) (Fig. 1f). Representative images of different imaging conditions are shown in Fig. 1h, where 53BP1 is labeled in red and BRCA1 is labeled in green (sample with Ku-60648 has relatively more green foci).

Fig. 1f **Fig. 1h**

Additionally, we used a complementary assay to determine the type of insertion-deletion mutations (indels) by Sanger sequencing after 3 days of Cas9 targeting ACTB in HEK293T cells. Exposure to Ku-60648 altered the indel outcomes at ACTB, from +1 insertions associated with NHEJ in favor of larger -3 deletions associated with MMEJ (Fig. 1j).

Fig. 1j

We identified comparable results using amplicon-NGS, this time at all on- and off-target sites for a gRNA evaluated by DISCOVER-Seq+. In K562 cells with Cas9 targeting FANCF site 2, we performed amplicon-NGS to evaluate editing efficiencies and indel outcomes (with vs without Ku-60648) at the on-target site and 14 off-target sites discovered by DISCOVER-Seq+.

Exposure to Ku-60648 led to a reduction in +/-1 indels from NHEJ in favor of larger -22 and -24 deletions from MMEJ (**Rebuttal Fig. 1**). We found that exposure to Ku-60648 led to 13 out of 15 target sites having detectable indels with amplicon-NGS (**Rebuttal Fig. 2**), compared to 7 out of 15 target sites with detectable indels without Ku-60648 (**Fig. 2k**). This may be explained by the fact that MMEJ is a highly error-prone repair pathway [PMID: 28754468, 28101326, 29804829, 33848455]; at off-target sites, especially those experiencing very limited Cas9 activity, having a high indel conversion rate such as with MMEJ is essential to obtain indel efficiencies above the detection limit. In contrast, NHEJ is generally more error-free due to direct ligation of broken DNA – previous studies by our group have estimated the conversion rate of NHEJ to be under 15% at a target site [PMID: 32527834].

We decided to not feature Rebuttal Fig. 1 or 2 in the manuscript for the following reason: amplicon sequencing was mainly employed in our study to determine if some off-target sites newly discovered with DISCOVER-Seq+ have evidence of mutations with standard CRISPR genome editing. The answer is yes, which is the focus of a paragraph in the Results section. We already know that DNA-PKcs inhibition changes repair outcomes, so also including amplicon sequencing data in cells with DNA-PKcs inhibition is not relevant to the main message and would only detract from that section. Furthermore, we already included a new paragraph early in the Results section that experimentally explores the effect of DNA-PKcs inhibition on DNA repair pathways, clearly showing modulation from NHEJ to MMEJ/HDR (**Fig. 1g-j**).

Rebuttal Fig. 1

Rebuttal Fig. 2

Fig. 2k

Together, these results confirm that DNA-PKcs inhibition with Ku-60648 blocks the NHEJ repair pathway in favor of MRE11-associated HDR/MMEJ pathways.

In contrast, we believe it is unlikely for DNA-PKcs inhibition to directly affect Cas9 behavior itself. Indeed, in the same U2OS STED imaging system, we see from Fig. 1g that the total number of DNA damage events after CRISPR activity is not significantly different with addition of Ku-60648, even though the DNA repair pathway has been modulated (Fig. 1f). The inhibitor Ku-60648 has also been used frequently in literature, either in the experimental setting or as a radiosensitizer for the potential treatment of cancer (<https://www.selleckchem.com/products/ku-0060648.html>). To our knowledge, there is no evidence that Ku-60648 impacts the function of the Cas9 protein itself. The concentration of inhibitor we use is also low at 1 uM in cells.

Fig. 1f-g

Together, our results indicate that DNA-PKcs inhibition does not affect Cas9 behavior, but affects DNA repair pathways. Most importantly for this reviewer's concern, these properties of Ku-60648 do not invalidate DISCOVER-Seq+ as an approach for off-target detection because the goal of DISCOVER-Seq+ is not to find mutagenesis but rather double strand breaks, which we argue is the more complete and relevant measure of CRISPR off-target activity.

4. It is necessary to provide the editing efficiencies (i.e., indel frequencies) at on-target or off-target sites in Figure 2d-related experiments.

*We thank the reviewer for raising this important topic. While measuring indel frequencies by amplicon-NGS in Cas9-treated mouse liver samples is feasible, we believe that it is relatively redundant in light of our new results in cell lines, and would not add significantly more to this manuscript. This is because as we showed in our revised manuscript, amplicon-NGS is a sub-optimal strategy for off-target detection. Because not all double strand cut events result in mutagenesis in a given experiment, it is more important to determine the totality of double strand cuts (the strength of DISCOVER-Seq+) rather than only those that produce mutagenesis. In agreement with this, in **Fig. 2k-l**, we showed there is a subset of off-target sites found by DISCOVER-Seq+ that did not have evidence of indels via amplicon-NGS. Additionally, we have new, strong evidence that confirms that these sites without indels are not false positive sites (**Fig. S2a-b**). Together, these results lead to the conclusion that all reported DISCOVER-Seq+ off-target sites are true positives, thereby confirming that amplicon-NGS is less sensitive since it fails to detect evidence of mutations at many off-target sites. Additional lines of evidence support this conclusion:*

- (4) A target site can experience DSBs but not lead to mutagenesis. Indeed, the rate of indel conversion after a DSB may not be high, estimated to be under 15% at a single site as shown in our recent study [PMID: 32527834].*
- (5) While it is important to identify mutagenesis, DSBs have deleterious effect even when adequately repaired and must therefore be identified. Indeed, recent evidence by our group showed that even a single DSB can inhibit cell division and induce local epigenetic changes such as increased accessibility [PMID: 36064968].*
- (6) Amplicon sequencing is critically underequipped to detect rare events. Amplicon sequencing does not enrich for Cas9-manipulated sites – assuming no amplification bias, amplicon-NGS reports on the ‘true’ mutation percentage at a given off-target site within its limit of detection, which can be approximated to be ~0.01% [PMID: 25513782]. By rough estimate, this means that in a mouse liver estimated to have approximately 300 million cells, more than 30,000 cells need to have a mutation at both alleles at a specific site to be detected by amplicon-NGS ($0.0001 * 3E8$), assuming adequate sequencing depth. In contrast, dedicated off-target detection methods such as DISCOVER-Seq+ specifically enrich for DNA fragments that are manipulated by CRISPR nucleases. For DISCOVER-Seq+, this is achieved by selectively amplifying MRE11-bound fragments; for GUIDE-seq, by selectively amplifying positions with double-strand oligodeoxynucleotide (dsODN) incorporated into genomic DNA; for BLISS/BLESS, by selectively amplifying positions adjacent to DSBs. These strategies specifically target CRISPR-manipulated sites, thereby increasing sensitivity while minimizing sequencing depth.*

Therefore, there can be a very low level of CRISPR activity at an off-target site that does not get detected by amplicon-NGS but is detected by DISCOVER-Seq+, suggesting that amplicon-NGS is not great at

detecting CRISPR off-target activity. Given that we have performed the complete set of amplicon-NGS for K562 FANCF site 2 and obtained interesting results presented in Fig. 2k-l and in response to a prior question, we believe that performing the same experiment on Cas9-treated mouse liver samples, while possible, is relatively redundant and would not add significantly more to this manuscript. We hope that the reviewer appreciates the new experiments and insights on these topics in our greatly expanded manuscript.

5. It would be better to discuss the limitation of the present method (DISCOVER-Seq+) in discussion part. *We thank the reviewer for this great suggestion. The main text now includes a Discussion section with a paragraph on the limitations of DISCOVER-Seq+.*

Decision Letter, first revision:

Dear Dr Ha,

Thank you for your letter asking us to reconsider our decision on your Brief Communication, "Improving the sensitivity of in vivo CRISPR off-target detection with DISCOVER-Seq+". After careful consideration we have decided that we are willing to consider a revised version of your manuscript that include the experimental data and analyses presented in your appeal rebuttal letter.

- * include a point-by-point response to our referees and to any editorial suggestions
- * please underline/highlight any additions to the text or areas with other significant changes to facilitate review of the revised manuscript
- * address the points listed described below to conform to our open science requirements
- * ensure it complies with our general format requirements as set out in our guide to authors at www.nature.com/naturemethods
- * resubmit all the necessary files electronically by using the link below to access your home page

[Redacted] This URL links to your confidential home page and associated information about manuscripts you may have submitted, or that you are reviewing for us. If you wish to forward this email to co-authors, please delete the link to your homepage.

We hope to receive your revised paper within 4 weeks. If you cannot send it within this time, please let us know. In this event, we will still be happy to reconsider your paper at a later date so long as nothing similar has been accepted for publication at Nature Methods or published elsewhere.

OPEN SCIENCE REQUIREMENTS

REPORTING SUMMARY AND EDITORIAL POLICY CHECKLISTS

When revising your manuscript, please submit reporting summary and editorial policy checklists.

IMAGE INTEGRITY

DATA AVAILABILITY

Please include a “Data availability” subsection in the Online Methods. This section should inform readers about the availability of the data used to support the conclusions of your study, including accession codes to public repositories, references to source data that may be published alongside the paper, unique identifiers such as URLs to data repository entries, or data set DOIs, and any other statement about data availability. At a minimum, you should include the following statement: “The data that support the findings of this study are available from the corresponding author upon request”, describing which data is available upon request and mentioning any restrictions on availability. If DOIs are provided, please include these in the Reference list (authors, title, publisher (repository name), identifier, year). For more guidance on how to write this section please see: <http://www.nature.com/authors/policies/data/data-availability-statements-data-citations.pdf>

CODE AVAILABILITY

Please include a “Code Availability” subsection in the Online Methods which details how your custom code is made available. Only in rare cases (where code is not central to the main conclusions of the paper) is the statement “available upon request” allowed (and reasons should be specified).

SUPPLEMENTARY PROTOCOL

To help facilitate reproducibility and uptake of your method, we ask you to prepare a step-by-step Supplementary Protocol for the method described in this paper. We [encourage authors to share their step-by-step experimental protocols](https://www.nature.com/nature-research/editorial-policies/reporting-standards#protocols) on a protocol sharing platform of their choice and report the protocol DOI in the reference list. Nature Portfolio's Protocol Exchange is a free-to-use and open resource for protocols; protocols deposited in Protocol Exchange are citable and can be linked from the published article. More details can found at a

href="https://www.nature.com/protocolexchange/about"
target="new">www.nature.com/protocolexchange/about.

ORCID

Nature Methods is committed to improving transparency in authorship. As part of our efforts in this direction, we are now requesting that all authors identified as 'corresponding author' on published papers create and link their Open Researcher and Contributor Identifier (ORCID) with their account on the Manuscript Tracking System (MTS), prior to acceptance. This applies to primary research papers only. ORCID helps the scientific community achieve unambiguous attribution of all scholarly contributions. You can create and link your ORCID from the home page of the MTS by clicking on 'Modify my Springer Nature account'. For more information please visit please visit www.springernature.com/orcid.

Best regards,
Lei

Author Rebuttal, second revision:

Reviewer #1:

Remarks to the Author:

In this revision, the manuscript has been strengthened with solid experiments. For the data, I have no further comments.

Reviewer #2:

Remarks to the Author:

The Authors have done a commendable work in addressing my comments and the other two Reviewers' remarks by performing several new experiments, most notably (1) the introduction of controls without Cas9 to assess whether the Ku-60648 alone induces DNA damage, and (2) the application of DISCOVER-seq+ to human CAR-T cells.

The Authors have also convincingly responded to my skepticism over the need of developing another method for OT detection and have expanded the Discussion section to reflect these considerations.

Reviewer #3:

Remarks to the Author:

The authors have mostly answered the issues I raised in the earlier review. I would like to recommend the publication of this revised version.

Response: We really appreciate the detailed comments and suggestions by all three reviewers, which greatly strengthened our manuscript. We are very thankful for the reviewers' strong support of our most recent submission.

Decision Letter, second revision:

29th Nov 2022

Dear Dr. Ha,

Thank you for submitting your revised manuscript "Improving the sensitivity of in vivo CRISPR off-target detection with DISCOVER-Seq+" (NMETH-BC48573B). It has now been seen by the original referees and their comments are below. The reviewers find that the paper has improved in revision, and therefore we'll be happy in principle to publish it in Nature Methods, pending minor revisions to satisfy any referees' final requests and to comply with our editorial and formatting guidelines.

TRANSPARENT PEER REVIEW

Nature Methods offers a transparent peer review option for new original research manuscripts submitted from 17th February 2021. We encourage increased transparency in peer review by publishing the reviewer comments, author rebuttal letters and editorial decision letters if the authors agree. Such peer review material is made available as a supplementary peer review file. Please state in the cover letter 'I wish to participate in transparent peer review' if you want to opt in, or 'I do not wish to participate in transparent peer review' if you don't. Failure to state your preference will result in delays in accepting your manuscript for publication.

Please note: we allow redactions to authors' rebuttal and reviewer comments in the interest of confidentiality. If you are concerned about the release of confidential data, please let us know specifically what information you would like to have removed. Please note that we cannot incorporate redactions for any other reasons. Reviewer names will be published in the peer review files if the reviewer signed the comments to authors, or if reviewers explicitly agree to release their name. For

more information, please refer to our [FAQ page](https://www.nature.com/documents/nr-transparent-peer-review.pdf).

ORCID

Sincerely,
Madhura

Madhura Mukhopadhyay, PhD
Senior Editor
Nature Methods

Reviewer #1 (Remarks to the Author):

In this revision, the manuscript has been strengthened with solid experiments. For the data, I have no further comments.

Reviewer #2 (Remarks to the Author):

The Authors have done a commendable work in addressing my comments and the other two Reviewers' remarks by performing several new experiments, most notably (1) the introduction of controls without Cas9 to assess whether the Ku-60648 alone induces DNA damage, and (2) the application of DISCOVER-seq+ to human CAR-T cells.

The Authors have also convincingly responded to my skepticism over the need of developing another method for OT detection and have expanded the Discussion section to reflect these considerations.

Reviewer #3 (Remarks to the Author):

The authors have mostly answered the issues I raised in the earlier review. I would like to recommend the publication of this revised version.

Final Decision Letter:

Dear TJ,

I am pleased to inform you that your Article, "Improving the sensitivity of in vivo CRISPR off-target detection with DISCOVER-Seq+", has now been accepted for publication in Nature Methods. Your paper is tentatively scheduled for publication in our May print issue, and will be published online prior to that. The received and accepted dates will be March 4, 2022 and March 6, 2023. This note is intended to let you know what to expect from us over the next month or so, and to let you know where to address any further questions.

Once your paper is typeset, you will receive an email with a link to choose the appropriate publishing options for your paper and our Author Services team will be in touch regarding any additional information that may be required.

Please note that *Nature Methods* is a Transformative Journal (TJ). Authors may publish their research with us through the traditional subscription access route or make their paper immediately open access through payment of an article-processing charge (APC). Authors will not be required to make a final decision about access to their article until it has been accepted. [Find out more about Transformative Journals](https://www.springernature.com/gp/open-research/transformative-journals)

Authors may need to take specific actions to achieve [compliance](https://www.springernature.com/gp/open-research/funding/policy-compliance-faqs) with funder and institutional open access mandates. If your research is supported by a funder that requires immediate open access (e.g. according to [Plan S principles](https://www.springernature.com/gp/open-research/plan-s-compliance)) then you should select the gold OA route, and we will direct you to the compliant route where possible.

For authors selecting the subscription publication route, the journal's standard licensing terms will need to be accepted, including [self-archiving policies](https://www.springernature.com/gp/open-research/policies/journal-policies). Those licensing terms will supersede any other terms that the author or any third party may assert apply to any version of the manuscript.

Your paper will now be copyedited to ensure that it conforms to Nature Methods style. Once proofs are generated, they will be sent to you electronically and you will be asked to send a corrected version within 24 hours. It is extremely important that you let us know now whether you will be difficult to contact over the next month. If this is the case, we ask that you send us the contact information (email, phone and fax) of someone who will be able to check the proofs and deal with any last-minute problems.

If, when you receive your proof, you cannot meet the deadline, please inform us at rjsproduction@springernature.com immediately.

Once your manuscript is typeset and you have completed the appropriate grant of rights, you will receive a link to your electronic proof via email with a request to make any corrections within 48 hours. If, when you receive your proof, you cannot meet this deadline, please inform us at rjsproduction@springernature.com immediately.

Once your paper has been scheduled for online publication, the Nature press office will be in touch to confirm the details.

Once your paper has been scheduled for online publication, the Nature press office will be in touch to confirm the details.

Content is published online weekly on Mondays and Thursdays, and the embargo is set at 16:00 London time (GMT)/11:00 am US Eastern time (EST) on the day of publication. If you need to know the exact publication date or when the news embargo will be lifted, please contact our press office after you have submitted your proof corrections. Now is the time to inform your Public Relations or Press Office about

your paper, as they might be interested in promoting its publication. This will allow them time to prepare an accurate and satisfactory press release. Include your manuscript tracking number NMETH-A48573C and the name of the journal, which they will need when they contact our office.

About one week before your paper is published online, we shall be distributing a press release to news organizations worldwide, which may include details of your work. We are happy for your institution or funding agency to prepare its own press release, but it must mention the embargo date and Nature Methods. Our Press Office will contact you closer to the time of publication, but if you or your Press Office have any inquiries in the meantime, please contact press@nature.com.

Nature Portfolio journals [encourage authors to share their step-by-step experimental protocols](https://www.nature.com/nature-research/editorial-policies/reporting-standards#protocols) on a protocol sharing platform of their choice. Nature Portfolio's Protocol Exchange is a free-to-use and open resource for protocols; protocols deposited in Protocol Exchange are citable and can be linked from the published article. More details can be found at www.nature.com/protocolexchange/about.

Best regards,
Madhura

Madhura Mukhopadhyay, PhD

Senior Editor
Nature Methods